

# Observations of VOC emissions and photochemical products over US oil- and gas-producing regions using high-resolution H₃O⁺ CIMS (PTR-ToF-MS)

Abigail Koss[1,2,3], Bin Yuan[1,2†], Carsten Warneke[1,2], Jessica B. Gilman[1], Brian M. Lerner[1,2*], Patrick R. Veres[1,2], Jeff Peischl[1,2], Scott Eilerman[1,2], Rob Wild[1,2], Steven S. Brown[1,3], Chelsea R. Thompson[1,2], Thomas Ryerson[1], Thomas Hanisco[4], Glenn M. Wolfe[4,5], Jason M. St. Clair[4,5], Mitchell Thayer[6], Frank N. Keutsch[6,7], Shane Murphy[8], Joost de Gouw[2,3]

1. NOAA Earth System Research Laboratory (ESRL), Chemical Sciences Division, Boulder, CO, USA
2. Cooperative Institute for Research in Environmental Sciences, University of Colorado Boulder, Boulder, CO, USA
3. Department of Chemistry and Biochemistry, University of Colorado Boulder, CO, USA
4. NASA Goddard Space Flight Center, Greenbelt, MD, USA
5. Joint Center for Earth Systems Technology, University of Maryland Baltimore County, Baltimore, MD, USA
6. University of Wisconsin Madison, Madison, WI, USA
7. Paulson School of Engineering and Applied Sciences and Department of Chemistry and Chemical Biology, Harvard University, Cambridge, MA, USA
8. University of Wyoming, Laramie, WY, USA
† now at Laboratory of Atmospheric Chemistry, Paul Scherrer Institute, 5232 Villigen, Switzerland
* now at Aerodyne Research, Inc., Billerica, MA, USA

Correspondence to: Carsten Warneke (Carsten.warneke@noaa.gov)

**Abstract.** VOCs related to oil and gas extraction operations in the United States were measured by H₃O⁺ chemical ionization time-of-flight mass spectrometry (H₃O⁺ ToF-CIMS / PTR-ToF-MS)

from aircraft during the SONGNEX campaign in March-April 2015. This work presents an overview of major VOC species measured in nine oil and gas producing regions, and a more detailed analysis of H₃O⁺ ToF-CIMS measurements in the Permian Basin within Texas and New Mexico. Mass spectra are dominated by small photochemically produced oxygenates, and compounds typically found in crude oil: aromatics, cyclic alkanes, and alkanes. Mixing ratios of

aromatics were frequently as high as those measured downwind of large urban areas. In the Permian, the H₃O⁺ ToF-CIMS measured a number of underexplored or previously unreported species, including aromatic and cycloalkane oxidation products, nitrogen heterocycles including pyrrole ($C_4H_5N$) and pyrroline ($C_4H_7N$), H₂S, and a diamondoid (adamantane) or unusual monoterpene. We additionally assess the specificity of a number of ion masses resulting from H₃O⁺

ion chemistry previously reported in the literature, including several new or alternate interpretations.



## 1. Introduction

Recent advances in fossil fuel extraction technology, especially horizontal drilling and hydraulic fracturing, have enabled a surge in crude oil and natural gas production in several regions across the United States over the past decade (US Energy Information Administration). A particular environmental concern is the release of air pollutants. Emissions can affect global climate, by the release of greenhouse gases (Miller et al., 2013;Brandt et al., 2014); regional air quality, by contributing ozone and particulate precursor species (Kemball-Cook et al., 2010;Edwards et al., 2014;McDuffie et al., 2016); and local air quality, by releasing air toxics harmful to human health (McKenzie et al., 2012;Adgate et al., 2014).

Detailed measurements of volatile organic compound (VOC) emissions, and their atmospheric reaction products, are needed to understand and mitigate these air quality concerns. Several studies have used gas chromatography (GC) techniques to characterize oil- and gas- related VOCs in relatively high chemical detail (Simpson et al., 2010;Gilman et al., 2013;Swarthout et al., 2013). These studies have demonstrated that comprehensive VOC characterization is invaluable for source identification and air quality modeling in these regions. To date, there are few such studies, and they have been limited in measurement of secondary species and in time resolution. The Permian Basin, located in west Texas and eastern New Mexico, is the physically largest and most productive oil field in the United States, but non-methane VOC measurements from this region have rarely been reported.

Proton-transfer-reaction mass spectrometry (PTR-MS) is a well-established chemical ionization technique used to measure VOCs, especially polar and aromatic species, in ambient air. This technique uses $H_3O^+$ reagent ions and can have time resolution of 1Hz or better. The recent development of PTR-MS instruments that use high- resolution time-of-flight mass analyzers (PTR-ToF-MS) has greatly expanded the number of measureable species, enhanced the technique's suitability to mobile platforms, and improved our ability to speciate specific ion masses (Jordan et al., 2009;Graus et al., 2010;Cappellin et al., 2011;Yuan et al., 2016a). For example, the resulting complex PTR-ToF-MS mass spectra in forested environments (Kim et al., 2010) and biomass burning (Brilli et al., 2014;Stockwell et al., 2015) have been reported. A recent study comparing PTR-ToF-MS and PTR-quadrupole MS instruments in an oil and gas producing



region in Utah pointed to the scientific advances possible with the application of PTR-ToF-MS, especially the separate measurement of hydrocarbon masses from isobaric oxygenates, and the measurement of previously undetectable photo-oxidation products (Warneke et al., 2015). In this work PTR-MS refers to the proton-transfer technique, and PTR-ToF-MS refers to PTR-MS instruments with a high-resolution time-of-flight mass analyzer. The instrument discussed in this work is called "$H_3O^+$ ToF-CIMS", an instrument similar to the PTR-ToF-MS but developed at NOAA.

This work comprises a detailed analysis of PTR-ToF-MS mass spectra obtained from measurements in oil- and gas-producing regions, supported by a comprehensive suite of other chemical instrumentation. We outline an interpretation of PTR-ToF-MS measurements in these regions, and report observed mixing ratios of commonly measured PTR-MS species in nine oil and gas producing regions. We present detailed VOC observations for the Permian Basin. This work provides detailed information about the VOC chemistry of these regions, will aid in the interpretation of PTR-MS (especially PTR-ToF-MS) measurements in oil and gas producing regions, and will support future analysis of ambient measurements in these regions.

## 2. Methods

### 2.1 Measurement location and context

Measurements were made from the NOAA WP-3D Orion research aircraft during the Shale Oil and Natural Gas Nexus (SONGNEX) campaign in March and April 2015. The SONGNEX campaign surveyed nine large oil and natural gas production regions in the central United States, several smaller producing regions, and locations with associated infrastructure. All research flights took place during daytime (late morning to mid-afternoon). The aircraft was equipped with a suite of chemical and meteorological instrumentation, which is described in section 2.2. This work reports measurements in nine regions: Bakken (ND), Upper Green River (WY), Uintah (UT), Denver-Julesburg (CO), San Juan (CO, NM), Permian (NM, TX), Barnett (TX), Eagle Ford (TX), and Haynesville (TX, LA) (Fig. 1a). SONGNEX measurements coincided with a peak in fossil fuel production in many of these regions, but were just after a downturn in the drilling of new wells due to a drop in the price of crude oil (Fig. 1b, 1c). Analysis is restricted to data collected in the





fossil fuel producing areas of the basins, and within the PBL (typically ≤ 600m AGL). Selection of data is shown in Fig. SI 3.

The Permian oil and gas field, located in western Texas and eastern New Mexico, was surveyed on three flights, on April 6, April 9, and April 23, 2015. Detailed interpretation of the $H_3O^+$ ToF-CIMS data focuses on measurements taken during the Permian flight on April 23. This flight featured high signal on many VOC masses, providing the best overall signal-to-noise ratio of any SONGNEX flight and allowing the observation of many VOCs that may have been below

detection limit on other flights. Additionally, there are few non-oil and gas emission sources in this region, which simplifies the interpretation of VOC measurements. During the April 23 flight, the average boundary layer temperature was 19 °C, the relative humidity ranged from 20-80% (average 34%), and wind speeds were typically between 2 and 10 m s$^{-1}$, averaging 5.4 m s$^{-1}$. The maximum concentration of ozone measured was 62 ppb, and the average 49 ppb.

The Permian Basin is an approximately 200 000 km$^2$ area encompassing a number of geologically distinct fossil fuel producing reservoirs, including several shale oil formations. The region is characterized by significant and intense oil production activity (accounting for nearly 20% of 2013 US domestic oil production), currently largely driven by recent development of shale oil

formations (Budzik and Perrin, 2014). As of September 2015, there were approximately 124 000 actively producing oil wells and 10 000 gas wells in this region, and approximately 1300 new wells were drilled in April 2015 (Railroad Commission of Texas, 2017;State of New Mexico Oil Conservation Division, 2017). Point sources reported in the NEI 2011 inventory are largely associated with oil and gas production, including, for example, refineries, processing facilities,

compressor stations, and tank batteries. The region has a population of 1 million, of whom about one-fifth live in the largest urban area, Midland-Odessa (U.S. Census). The climate is arid and the land cover consists mainly of desert and grassland. The April 23 SONGNEX flight track; locations of oil and gas wells, urban areas, and NEI 2011 point sources, and the spatial distributions of a few VOCs of interest are shown in Figure 2.

To support the interpretation of SONGNEX $H_3O^+$ ToF-CIMS measurements, this work also refers to measurements taken by an Ionicon PTR-ToF-MS instrument in the Uintah Basin, Utah, oil and



gas field during winter of 2013. The Ionicon PTR-ToF-MS instrument relies on the same measurement principle as the SONGNEX $H_3O^+$ ToF-CIMS and has similar mass resolution. The PTR-ToF-MS Uintah Basin measurements have been previously described by Warneke et al. (2015), and a comparison between the $H_3O^+$ ToF-CIMS and PTR-ToF-MS instruments is given by Yuan et al. (2016a). Finally, we include $H_3O^+$ ToF-CIMS headspace measurements (at 25 °C) of a crude oil sample. The sample was purchased from ONTA (Geology) Goal and Petroleum Inc. and is a blend of oil from several reservoirs in West Texas. The sample was bottled at a central collection facility after transport in tanker trucks and before being sent to a refinery. It is possible that some VOCs were removed prior to transport: sulfur and nitrogen-containing organics are often removed at processing facilities, although we do not know how this sample was treated at the collection facility. Some of the most highly-volatile VOCs may have been depleted during transport.

## 2.2 Instrumentation

### 2.2.1 Description of $H_3O^+$ ToF-CIMS instrument

The $H_3O^+$ ToF-CIMS and its operation during the SONGNEX campaign have been previously reported (Yuan et al., 2016a). The basic operational principle is the same as other PTR-MS instruments. $H_3O^+$ ions are generated from water vapor in a hollow cathode discharge ion source. The $H_3O^+$ reagent ions are then mixed with ambient air, containing VOCs, in a drift tube section. The proton from $H_3O^+$ is transferred to VOCs with sufficiently high proton affinity, and the resulting ionized VOCs are transferred to a mass analyzer (in the $H_3O^+$ ToF-CIMS, VOC ions are guided through a set of RF-only segmented quadrupoles to a time-of-flight unit). The hollow cathode, drift tube, pressure controlled inlet, background, and calibration components are custom built; the ion transmission and time-of-flight analyzer were produced by Aerodyne Research Inc./Tofwerk. The $H_3O^+$ ToF-CIMS has a drift tube E/N (electric field strength divided by number density) of about $120 \times 10^{-17}$ Td (V cm$^2$). Data are presented in this work at 1s time resolution unless otherwise noted.

During field operation, the instrument background was determined for 90 seconds every 20-40 minutes by flowing ambient air through a catalyst, and a 10-component gas standard was added





for 90 seconds every 1-2 hours (single-point calibration) to record instrument stability and sensitivity. In addition to the in-flight single-point calibrations, multi-point calibrations were performed at the beginning and end of each flight. A small amount of trichlorobenzene ($C_6H_3Cl_3$)

was continuously introduced into the instrument as a mass calibrant. Data were analyzed using Tofware high resolution peak-fitting software (Aerodyne Research Inc./Tofwerk). Data were then corrected for humidity-dependent sensitivities and background-subtracted. In addition to the in-flight calibrations, laboratory calibrations of a larger number of species were performed using standard cylinders and permeation tubes. More details on the data quality assurance procedures

related to instrument operation, background subtraction, humidity dependence, and calibration are included in Yuan et al. (2016a).

The $H_3O^+$ ToF-CIMS instrument has a mass resolution of approximately 3900-5900 m/$\Delta$m over the mass range discussed in this work (m/z 12-200), which is sufficient to determine the elemental

formulas of most detected ion masses. Evaluation of data quality related to high-resolution peak-fitting are discussed in Supplementary Information (Section SI 1). Also included in the Supplementary Information are an extensive set of high-resolution mass spectra showing isobaric contributions to nominal masses, which will be useful to operators of both unit-mass- and high-resolution PTR-MS instruments (Fig. SI 4).

In this work we report signal intensity using normalized counts-per-second (ncps), and VOC mixing ratio (ppbv) for calibrated species. Normalized counts-per-second (ncps) is the instrument signal relative to $10^6$ $H_3O^+$ ion counts, corrected for humidity effects, and background subtracted. VOC sensitivities were determined by (1) direct calibration, where available; (2) calculated using

proton-transfer rate constants, either known or calculated based on polarizability and dipole moment (Sekimoto et al., 2017); or (3) an average sensitivity determined from the calibrated and calculated sensitivities. Accuracy is within 15% for directly calibrated compounds and generally within a factor of two for calculated sensitivities. Some ion masses, such as m/z 45.992 $NO_2^+$ (Section 3.4.4) and m/z 81.070 $C_6H_8H^+$ (Section 3.4.3) have an ambiguous interpretation and these

are discussed in terms of instrument signal (ncps) and not mixing ratio.





Measured 1-s detection limits (for a signal-to-noise ratio of 3) range from approximately 40 pptv (aromatics) to 400 pptv (methanol). Some mixing ratios reported in this work are smaller than 40 pptv. In a few cases, the signal-to-noise ratio is less than three, but variability is still discernable, and these species are discussed mostly for the *absence* of significant enhancement (e.g. styrene, cresol). Other species are presented as an average over a longer period of time: for example, average boundary layer enhancements presented in Figure 5, and time series in Figure 13. As averaging time increases, the limit of detection decreases: a typical aromatic compound with 1s detection limit of 40 pptv (sensitivity of 500 count ppbv$^{-1}$ s$^{-1}$, background of 5 counts per second) has a 10s detection limit of about 9 pptv (a calculation is included in the supplementary information, Section SI 2).

### 2.2.2 PTR-MS application to oil and gas emissions: strengths and limitations

In the WP-3D SONGNEX payload, the primary strengths of the H$_3$O$^+$ ToF-CIMS include measurement of small acids and carbonyls not detected by other instruments, and a much higher measurement rate of aromatics and cyclic alkanes, which were also measured by the whole air sampler, which typically collected sample for five seconds once every three minutes (Lerner et al., 2017). Aromatics and compounds with heteroatoms are generally detected sensitively (aromatics: measured average 180 ncps/ppb, polar compounds: 200 ncps/ppb) with a few exceptions of compounds that can dissociate by dehydration, such as small alcohols and aldehydes (de Gouw and Warneke, 2007). Cyclopentane and cyclohexane cannot be detected, but alkyl-substituted cyclic alkanes are detected, at approximately 5% of the sensitivity of aromatics. Alkenes containing four or more carbon atoms are detected sensitively (estimated average 300 ncps/ppb), although larger alkenes can fragment at the high E/N conditions used in our instrument (Gueneron et al., 2015).

PTR-MS is notably limited in its measurement of alkanes: saturated alkanes smaller than hexane have too low proton affinity to be detected, and C6 and larger branched-and straight-chain alkanes are detected with very low sensitivity, reacting with H$_3$O$^+$ at a rate one or two orders of magnitude slower than aromatics (Arnold et al., 1998). Additionally, the larger alkanes fragment extensively (Gueneron et al., 2015). This behavior makes alkanes difficult to measure sensitively and selectively. This is particularly limiting in measurements of emissions from oil and gas operations,





where alkanes are typically dominant in terms of mixing ratio. $NO_2^+$ cannot be converted to a mixing ratio in a meaningful way and was excluded from the concentration comparison. For hydrocarbons, an average sensitivity was applied (Section 2.2.1).

During SONGNEX, fast measurement of methane and ethane were provided by cavity ring-down and direct-absorption spectroscopy (respectively), and speciated C2-C8 alkanes were measured by whole-air-sampling/GC-MS. Data from a number of other chemical instruments that were deployed during the SONGNEX mission are used in this analysis. Descriptions of these

225 instruments are given in Table 1. Further information can be found at https://esrl.noaa.gov/csd/groups/csd7/measurements/2015songnex/P3/datainfo.html.

## 3. Results and Discussion

### 3.1 Overview of $H_3O^+$ ToF-CIMS measurements

In this section, we provide a comparison of the mixing ratios of several volatile species between

230 all flights made in the San Juan, Uintah, Upper Green River, Denver-Julesburg, Barnett, Permian, Haynesville, Bakken, and Eagle Ford regions. Species measured by $H_3O^+$ ToF-CIMS that are included in this comparison are the sum of C6-C10 aromatics, methanol, $H_2S$, and acetone. Figure 3 shows an overview of the mixing ratios of these compounds in the boundary layer during each flight. Additionally, monoterpenes (measured by iWAS/GC-MS), $NO_x$ (measured by the

235 chemiluminescence instrument), $CH_4$ (measured by cavity ring-down spectroscopy), and average wind speed within the boundary layer are shown in Figure 3 for chemical and meteorological context.

BTEX species (benzene, toluene, and C8 aromatics), and higher weight aromatics, are important

air toxics and ozone and aerosol precursors that are commonly reported in VOC measurements of oil and gas-producing regions. PTR-MS measures these compounds with high specificity and sensitivity (de Gouw and Warneke, 2007). Figure 3a shows that aromatics concentrations in oil and gas basins were typically high – comparable to concentrations downwind of large urban areas.




The $H_3O^+$ ToF-CIMS measurement of benzene, toluene, and C8 aromatics agrees well with measurements made by whole-air-sampling followed by GCMS analysis (iWAS/GC-MS) (Yuan et al., 2016a;Lerner et al., 2017). There were large differences in observed mixing ratios between basins. On all flights, sharp, concentrated plumes of aromatics were encountered (resulting in maximum and average mixing ratio much higher than median). Large differences in mixing ratio between different flights in the same basin (e.g. Uintah, Denver-Julesburg) are partially the result of differences in meteorological conditions while in other basins (Permian, Haynesville, Upper Green River, Eagle Ford, Bakken), average mixing ratios were more consistent between flights. Benzene was not measured by the $H_3O^+$ ToF-CIMS during the March 24 San Juan flight due to an instrument issue. For comparison, Figure 3 also shows boundary layer statistics from two flights during other recent P3 aircraft campaigns: SENEX (June 16 2013 flight, Atlanta GA metropolitan area) and CALNEX (May 05 2010 flight, Los Angeles CA metropolitan area). SENEX and CALNEX data are from a PTR-quadrupole MS instrument. The maximum mixing ratios of aromatics during every SONGNEX flight were considerably higher than those measured during SENEX and CALNEX (note the log scale); averages within several basins (especially Uintah, Denver-Julesburg, and Permian) were comparable to or higher than aromatics measured over the Los Angeles metropolitan area (population of 18 million).

$H_2S$ is an air toxic that can be emitted from oil and gas sources, and can seriously harm human health (Tarver and Dasgupta, 1997;Li et al., 2014). Significant enhancements of $H_2S$ were seen only in the Permian and Haynesville regions (Fig. 3c). All three Permian flights saw broad enhancements in $H_2S$ likely associated with oil and gas production. Emissions from a paper mill were captured during the Haynesville flights and a mixing ratio of 27.7 ppbv was measured 20.5 km downwind of the point source, which was the highest $H_2S$ mixing ratio measured during SONGNEX. The Eagle Ford flights had a higher limit-of-detection due to an unknown instrument issue and the maximum mixing ratio is not statistically significant.

Oxygenated compounds provide insight into photochemical aging, and comprised the majority of $H_3O^+$ ToF-CIMS product ion signal. The most abundant oxygenated compounds were methanol and small (C1- C4) carbonyls and acids. These compounds include m/z 45.034 $C_2H_4OH^+$ (acetaldehyde), m/z 59.049 $C_3H_6OH^+$ (acetone), m/z 73.065 $C_4H_8OH^+$ (2-butanone), m/z 61.028





$C_2H_4O_2H^+$ (acetic acid), and m/z 75.044 $C_3H_6O_2H^+$ (propionic acid). Support for our interpretation of these ion masses is provided in Section 3.4.1.

Mixing ratios of acetone were similar in each basin, with average mixing ratios within a range of about 1.5 ppb (Fig. 3b). Mixing ratios of acetaldehyde, MEK, and acetic acid for each flight are in Figure SI 5. The relative abundances of oxygenates were generally similar between basins (Fig. 4), although there was higher variability in organic acids and formaldehyde. The highest relative abundances of organic acids were observed in the Denver-Julesburg Basin and are likely due to primary emissions from concentrated animal feeding operations and not photochemistry (Eilerman 285 et al., 2016;Yuan et al., 2017).

Figure 4 compares the distribution of oxygenates to that measured during the UBWOS 2013 campaign (Utah Division of Air Quality, 2014), the SENEX and CALNEX flights, and the CALNEX ground site in Los Angeles (Veres et al., 2011;Warneke et al., 2013). The mix of VOC 290 precursors from oil and gas fields (dominated by small alkanes) is very different from urban and biogenically influenced air. However, the mix of products was similar to that measured during CALNEX. The similarity between the SONGNEX and CALNEX profiles is somewhat surprising since it has been shown that the oxidation mechanisms in urban and oil and gas producing regions are quite different (Yuan et al., 2015). The distribution of carbonyls was similar to that measured 295 during the SENEX flight, but had significantly lower formaldehyde and formic acid, likely due to the much lower concentration of their precursor, isoprene.

Methanol (detected at m/z 33.034 $CH_4OH^+$) was the single most abundant VOC detected by the $H_3O^+$ ToF-CIMS. There is agreement to within the stated uncertainties between the $H_3O^+$ ToF-300 CIMS measurement and the iWAS/GC-MS measurement ($R^2$= 0.9, slope = 1.23), despite the difficulty in retrieving methanol from the whole-air-sampling system (Lerner et al., 2017). There was high variability in methanol mixing ratios between basins, between flights within the same basin, and within individual flights (Fig. 3d). Also shown, in Figure 3f, is the iWAS/GC-MS measurement of monoterpenes, as a proxy for biogenic emissions.




The two flights with the highest methanol concentrations were the April 25 Haynesville flight, and the April 23 Permian flight, both of which had mixing ratios comparable to those observed during SENEX, a summertime campaign over a biogenically productive region. All three Permian flights had high methanol mixing ratios.


To summarize, oil and gas-producing regions, even those in rural areas such as the Permian and Uintah basins, can have VOC mixing ratios comparable to those measured in urban areas. The concentrations and relative distribution of photochemically produced species were relatively similar between basins. There are significant differences between the basins in the mixing ratios

and composition of primary compounds such as aromatics and $H_2S$. Forthcoming work, including measurements from iWAS/GC-MS, will investigate these differences and their origins in greater detail.

### 3.2 Overview of measurements in the Permian Basin

The highest overall mixing ratios of VOCs were detected during the April 23 flight over the Permian Basin. In the following sections we provide a detailed interpretation of PTR-ToF-MS mass spectra in oil and gas producing regions, based on observations in the Permian Basin.

An averaged mass spectrum from the boundary layer in the Permian Basin flight on April 23 is

shown in Figure 5a, where all peaks shown are well above the detection limit. Enhancements are relative to the average mixing ratio in a ten-minute free-troposphere measurement immediately prior to descent into the basin. (The time period and altitude of this measurement are shown in Figure SI 3).

Compounds containing one or two oxygen atoms dominate the product ions measured by the $H_3O^+$ ToF-CIMS. These oxygenates are comprised mainly of methanol (m/z 33.034 $CH_4OH^+$) and small photochemical products. Other important ions include masses typically attributed to aromatics and cyclic alkanes; m/z 34.995 $H_2S \cdot H^+$ (hydrogen sulfide); and m/z 45.992 $NO_2^+$. Figure 5b compares the average boundary layer concentrations of species detected by $H_3O^+$ ToF-CIMS. $NO_2^+$ cannot

be converted to mixing ratio in a meaningful way and was excluded from the concentration



comparison. For fragmentary hydrocarbons, an average sensitivity was applied (Section 2.2.1). It can be seen that methanol and $H_2S$, while comprising a relatively small portion of the overall signal, are actually quite important in terms of actual abundance.

The relationship between major $H_3O^+$ ToF-CIMS product ion signals measured over the Permian and the West Texas crude oil headspace sample is shown in Figure 6. Photochemical products like the small oxygenates, and methanol, are greatly enhanced compared to hydrocarbon masses, which are generally similar to the composition of crude oil. Alkane and alkene masses are somewhat more abundant relative to aromatics in the SONGNEX measurements, compared to the crude oil.

This could be due to compositional differences between crude oil and VOC emission sources in the Permian, additional signal on alkane masses from photochemical products (Section 3.2.4), or photochemical removal of aromatics (Section 3.3.2). A few species, particularly $H_2S$, were enhanced in the Permian but not detected in the crude oil. There are large differences in the $H_2S$ content of various Permian basin reservoirs (Railroad Commission of Texas) and it is possible that

our crude oil sample was derived from low-sulfur-content reservoirs, or that sulfur species were removed prior to bottling.

During the April 23 flight, several compositionally distinct air masses were sampled (Fig. 2). One air mass, in the southwestern part of the flight path, was enriched in aromatics, methane, ethane,

and other primary compounds (less aged); another, to the east, was relatively more enriched in oxygenates that are typically photochemically produced, such as acetone, acetaldehyde, and PAN species (more aged). HYSPLIT back-trajectory modeling indicates that over the 24 hours prior to sampling, the more aged air mass circulated over the southwestern part of the oil field. The highest concentrations of acetone and acetaldehyde occur over the topographically lower areas of the

flight. A reasonable explanation for the more aged air mass is that it consists of emissions and chemical products from the previous day pooled in low-lying areas. Most VOCs are enhanced in the less-aged spatial distribution (e.g. toluene), the more-aged spatial distribution (e.g. acetaldehyde), or a combination of the two (e.g. m/z 83.086 $C_6H_{10}H^+$). Figure 7 compares the correlation of the measured VOCs with toluene, a primary emission, and acetaldehyde, a secondary

product. Alkanes, aromatics and cycloalkanes have a higher correlation with toluene, while oxygenates have a higher correlation with acetaldehyde. A small number of species, such as $H_2S$


and m/z 71.049 $C_4H_6OH^+$, have different distributions. The physical separation between less and more aged emissions during this flight was used to help identify VOC ions and to interpret their source.


The three transects in the western part of the flight path contain similar VOC composition, but the concentrations of toluene, C8 aromatics, and larger aromatics decrease relative to benzene from south to north. The relative decreases of the more reactive aromatics is likely the result of longer photochemical processing, which is consistent with the southerly wind direction during the flight,

generally lower concentrations from south to north, and the time of each transect (northernmost transect latest in the day). We used the ratios of toluene, C8 aromatics, and C9 aromatics to benzene to calculate a relative OH exposure for each transect, in order to explore photochemical aging of other VOCs. This method has been used extensively in atmospheric chemistry (de Gouw et al., 2005;Warneke et al., 2013) and details are shown in the supplementary information (SI Section 3).

Not all species had significant enhancement in this area of the flight (e.g. $H_2S$ had a different spatial distribution), so we were not able to use this method to investigate photochemical aging these species.

In the following discussion, we examine several groups of compounds measured during the April 23 Permian fligth in greater detail: hydrocarbons (aromatic, cyclic alkane, alkene, and alkane

masses), secondary compounds, and compounds with heteroatoms. Table 2 lists ion masses discussed in this work and our interpretation of that measurement. Table 2 also highlights (in bold text) measurements that provide new understanding of atmospheric composition and chemistry: VOCs that have been previously unreported or underexplored, and VOC ion masses where our assessment is a new interpretation of a mass previously reported in the literature.

**3.3 Hydrocarbon masses**

**3.3.1 Aromatics**

Alkyl-substituted aromatic species are commonly measured by PTR-MS and the interpretation of these ion masses is well established (de Gouw and Warneke, 2007;Blake et al., 2009). Aromatics can be important to the photochemical production of ozone in oil and gas producing regions, and

can have high yields of secondary organic aerosol (Henze et al., 2008;Edwards et al., 2014).





Quantifying the distribution of these species emitted from oil and gas sources is important to modeling work and to distinguishing between oil and gas and urban sources.

Figure 8 summarizes the distribution of C7 C10 alkyl-substituted aromatics relative to benzene measured during SONGNEX and compares the findings with other sources. This figure compares (1) measurements from all flights, (2) the average aromatics enhancement over the Permian during the April 23 flight, (3) fresh emissions sampled in a plume from a point source (a natural gas processing plant, 32.49N 101.35W) on the April 6 and April 9 Permian flights, (4) aromatics in the headspace of West Texas crude oil, (5) a literature survey of oil and gas sources, and (6) a literature survey of urban sources. The profiles of the average for all SONGNEX flights, and of aromatics measured in the Permian, are similar to the profile typically measured in oil and gas producing regions. The Permian profile is very similar to the composition of West Texas crude oil, and there is significant diversity in aromatics profiles between different oil and gas regions and sources. The average oil and gas profile is clearly differentiated from urban emissions (vehicular sources) by roughly equal enhancements of toluene and benzene, followed by a steady decrease in enhancement with increasing carbon number. The average oil and gas, and urban composition shown in Figure 8d were calculated by averaging all oil and gas sources (panels a and b) and all literature profiles shown in panel 8c for the urban profile.

The $H_3O^+$ ToF-CIMS also allowed for the investigation of more highly alkyl-substituted aromatics with 4 double bond equivalents; and less saturated aromatics, including PAHs and styrenes. During the April 23 Permian flight, we found that benzene and C7-C9 alkyl substituted aromatics were by far the most abundant aromatic species, accounting for 95% of total aromatic signal. These species were observed in both broad enhancements and in localized plumes from point sources. No PAHs with significant enhancement (above estimated 1s detection limit of 30-40 pptv) were observed. Over all flights, styrene (m/z 105.070 $C_8H_8H^+$) was consistently enhanced by up to 1.1 ppbv in plumes from point sources, and up to 60 pptv during one leg of the April 6 flight, where it was not correlated with other aromatics.



### 3.3.2 Cycloalkanes

Cycloalkanes are an important component of crude oil (Smith, 1968;National Research Council, 1985;Drozd et al., 2015), and have been detected by GC in significant concentrations in the atmosphere over oil and gas producing regions (Simpson et al., 2010;Gilman et al., 2013;Edwards et al., 2014). Although cycloalkanes are measured by PTR-MS with lower sensitivity than aromatics, the resulting product ions are still detectable. We find that cycloalkane product ions are

not specific for cycloalkanes, but may still be useful for characterizing the VOC composition of air masses, and may be specific for cycloalkanes in relatively unaged air masses.

Several previous laboratory and field experiments have explored the PTR-MS response to cycloalkanes. PTR-MS is somewhat less sensitive to cyclic alkanes than to aromatics and

oxygenates (Midey et al., 2003;Gueneron et al., 2015). At the E/N conditions in our instrument, cycloalkanes experience significant fragmentation, creating important product ions at m/z 69.070 $C_5H_9^+$, 83.086 $C_6H_{11}^+$, 85.101 $C_6H_{13}^+$, 97.101 $C_7H_{13}^+$, 111.117 $C_8H_{15}^+$, and 125.132 $C_9H_{17}^+$ (Midey et al., 2003;Warneke et al., 2003;Yuan et al., 2014;Gueneron et al., 2015). Cyclic alkanes have been measured with PTR-MS in crude oil using a GC interface (Yuan et al., 2014), and in ambient

air in the Uintah basin (Warneke et al., 2014); based on these measurements we expect that m/z 69.070 is produced generally by C6-C9 cycloalkanes; m/z 83.086 mainly from methylcyclopentane, cyclohexane, and methylcyclohexane; m/z 85.101 mainly by methylcyclopentane; m/z 97.101 mainly by methylcyclohexane; and m/z 111.117 and 125.132 by C9 and C10 cycloalkanes, respectively. These masses were clearly enhanced in $H_3O^+$ ToF-CIMS

measurements over the Permian and other basins. Cumulatively, they account for 50% of the hydrocarbon concentration (ppbv) measured by $H_3O^+$ ToF-CIMS.

Chemically specific measurements of methylcyclohexane and cyclohexane were made by iWAS/GC-MS. The comparison between the $H_3O^+$ ToF-CIMS and iWAS/GC-MS

methylcyclohexane measurements shows that the $H_3O^+$ ToF-CIMS measurement is on average three times higher and the correlation coefficient $R^2$ is 0.71. The ratio of m/z 97.101 to iWAS methylcyclohexane is higher in the more aged air mass (Fig. 9a). This suggests that m/z 97.101 includes a significant contribution from a secondary VOC that fragments to produce $C_7H_{13}^+$. Some

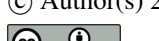



possibilities include an aldehyde or alcohol with formula m/z 115.112 $C_7H_{14}OH^+$ (e.g. heptanal or

methylcyclohexanol) or a larger molecule (e.g. m/z 129.127 $C_8H_{16}OH^+$ losing $-CH_3OH$).

Comparisons of methylcyclohexane between other GC and PTR-MS instruments previously deployed in the Uintah Basin, Utah, indicate that this behavior is not an $H_3O^+$ ToF-CIMS specific issue and is not unique to the Permian Basin. In the winter of 2012, when photochemistry was

low, quadrupole PTR-MS m/z 97 measurements agreed with the methylcyclohexane measurement from a GC-MS instrument within 23% (Warneke et al., 2014). However, in the winter of 2013, when photochemistry was much higher (Edwards et al., 2014), the relationship between PTR-ToF-MS m/z 97.101 $C_7H_{13}^+$ and GC-FID methylcyclohexane was dependent on photochemical exposure (Fig. 9b).


The relatively high correlations of other cycloalkane indicator masses (m/z 83.086, m/z 69.070, m/z 111.117, etc.) with acetaldehyde suggest that they too experience interference from similar secondary products (Fig. 7). The secondary compounds could be cycloalkane oxidation products, or other alcohols or aldehydes, which fragment by losing a water molecule -- a common PTR-MS

fragmentation mechanism (Yuan et al., 2016a).

### 3.3.3 Interpretation of hydrocarbon ion masses in oil and gas regions

The $H_3O^+$ ToF-CIMS measurements during SONGNEX offer new insights into some VOC ion masses commonly measured by PTR-MS.

It is clear from the comparison to iWAS/GC-MS measurement of isoprene (Fig. 10) that m/z 69.070 $C_5H_9^+$ is not isoprene, which is the dominant contributor in many air masses (de Gouw and Warneke, 2007;Blake et al., 2009). We interpret it as the sum of a cycloalkane fragment and a secondary compound from the oxidation of oil and gas precursor emissions; several other studies have also suggested non-biogenic interpretations of this mass (Yuan et al., 2014;Gueneron et al.,

2015). Similarly, m/z 137.132 $C_{10}H_{16}H^+$, usually interpreted as monoterpenes, was enhanced far above the monoterpene species (α- and β-pinene) detected by iWAS/GC-MS. M/z 137.132 $C_{10}H_{16}H^+$ was also observed using a PTR-ToF-MS instrument in the Uintah Basin, UT in winter of 2013 (Fig. SI 6). The Uintah observations indicate that this unknown species is emitted in





several oilfields and is likely not an instrument artifact. Possible alternative sources of this VOC
ion mass are (1) another isomer of $C_{10}H_{16}$, but not a monoterpene; (2) a monoterpene species not
detected by iWAS/GC-MS; or (3) another species, such as an alcohol or aldehyde, which
fragments to m/z 137.132. Each possibility is discussed in more detail below.

(1) *Other $C_{10}H_{16}$ isomer.* Other, non-terpene isomers of $C_{10}H_{16}$ are found in fossil fuels. In
particular, the presence of "diamondoids", characterized by cage-like structures, is well known
(Dahl et al., 1999;Araujo et al., 2012). The smallest diamondoid molecule, adamantane
(tricyclo[3,3,1,1(3,7)]decane, $C_{10}H_{16}$), has a tetrahedral structure formed by three cyclohexane
rings (Fig. 10). Adamantane can have concentrations from <1% to 100% of typical benzene
concentrations in crude oil (Verma and Tombe, 2002;Araujo et al., 2012). In the Permian
atmospheric measurement, m/z 137.132 $C_{10}H_{16}H^+$ was up to 20% the concentration of benzene
(assuming similar sensitivity as monoterpenes). Alkyl-substituted adamantanes are relatively
abundant (Stout and Douglas, 2004;Wang et al., 2006). We detected enhancement of m/z 151.148
$C_{11}H_{18}H^+$, but larger molecules were below the detection limit of the $H_3O^+$ ToF-CIMS during this
flight. The measured headspace of West Texas Crude showed relative enhancements of $n(CH_2)$-
substituted $C_{10}H_{16}$ that are more consistent with the adamantane series than with terpenes, where
we would expect to see enhancements at every $n(C_5H_8)$, but the evidence is not conclusive (Fig.
SI 7). Adamantane has a longer retention time than the GC elution time used for the iWAS samples,
so iWAS/GC-MS was not able to confirm or exclude the presence of this molecule.

(2) *Other monoterpene.* Terpene-derived ("isoprenoid") saturated compounds are known to be in
crude oil, and some larger molecules such as phytane and pristane are used as geochemical
biomarkers (Tissot and Welte, 1984). There is less information available about the presence of
unmodified monoterpenes, but the presence of an unusual monoterpene (such as limonene) derived
from crude oil or industry solvents is certainly possible. Data from the iWAS/GC-MS was again
inconclusive, but suggested that it was not limonene.

(3) *Fragment of another species.* M/z 137.132 $C_{10}H_{16}H^+$ is not likely to be a fragment of another
species. We did not detect any oxygenated species that could easily fragment to produce m/z 137.
A larger hydrocarbon could fragment to this mass. However, in crude fuels, hydrocarbon





concentrations in a particular homologous series generally decrease with carbon number. An anomalously intense larger mass would require its own explanation.

We do not currently have enough information to state conclusively if the observed m/z 137.132 $C_{10}H_{16}H^+$ ion is from adamantane, a monoterpene, or another isomer. Collection of more evidence

is outside the scope of this manuscript and is currently under separate investigation. If the observed $C_{10}H_{16}$ molecule is indeed adamantane, it represents a previously undetected class of atmospheric VOCs. Regardless of identity, this observation highlights that there may be significant emissions of higher-mass hydrocarbons from fossil fuels that are rarely measured. These compounds could be used to identify emission sources, and could potentially be SOA precursors (de Gouw et al.,

525    2011).

### 3.3.4 Other hydrocarbon masses

The $H_3O^+$ ToF-CIMS also measured other hydrocarbon signals that could not be tied to a single VOC, and these hydrocarbon masses comprise a significant fraction of the total hydrocarbon signal (Fig. 5). These other important hydrocarbon ion masses include m/z 41.039 $C_3H_4H^+$, m/z 43.054

$C_3H_6H^+$, m/z 57.070 $C_4H_6H^+$, and m/z 71.086 $C_5H_{10}H^+$. Unit-mass-resolution PTR-MS has not been able to investigate these ions because of strong interference from isobaric masses, such as $C_2H_2OH^+$ at m/z 43.018. Several studies have reported that these masses are non-specific and are produced by alkenes, and fragmentation of alkanes, alkenes, aldehydes, and alcohols (Buhr et al., 2002;Jobson et al., 2005;Gueneron et al., 2015). Laboratory tests with the $H_3O^+$ ToF-CIMS

confirmed that multiple alkane and alkene VOCs present during the April 23 flight (as measured by iWAS/GC-MS) fragment to these masses. These ion masses had significant intensity in both the southwestern (less aged) and eastern (photochemically enhanced) areas of the flight, suggesting contributions from fragments of both primary and secondary species.

### 3.4 Hydrocarbon oxidation chemistry

### 3.4.1 Major secondary compounds

Our interpretation of the most abundant photochemically produced ion masses is consistent with interpretations often presented in the literature: m/z 45.034 $C_2H_4OH^+$ is specific for acetaldehyde,




m/z 59.049 $C_3H_6OH^+$ for acetone, m/z 73.065 for MEK, m/z 61.028 $C_2H_4O_2H^+$ for acetic acid, and m/z 75.044 $C_3H_6O_2H^+$ for propionic acid.

The carbonyl species with more than three carbon atoms have both ketone and aldehyde isomers, and both isomers could be products of alkane oxidation. For example, oxidation of propane in the presence of $NO_x$ is expected to yield approximately 26% propanal and 74% acetone (Calvert et al., 2008). The ratio of propanal to acetone has been measured to be 0.17 (Uintah Basin, 2012, Edwards et al. (2013)), 0.19 (Uintah Basin, 2014), and 0.22 (Denver-Julesburg basin, Gilman et al. (2013)) in oil and gas producing regions, compared to 0.06 in an urban area (Los Angeles, Borbon et al. (2013)). However, almost all of the signal at $H_3O^+$ ToF-CIMS m/z 59.049 $C_3H_6OH^+$ can be attributed to acetone. The PTR-MS reaction of $H_3O^+$ with propanal is dissociative, and the sensitivity of the $H_3O^+$ ToF-CIMS to propanal at m/z 59.049 is only 3% that of acetone. Acetone therefore dominates the signal at this mass, and we assume similar behavior for higher-mass carbonyls.

Abundances of larger masses in the 1-double bond equivalent, 1-oxygen homologous series decrease rapidly with carbon number; these are most likely also carbonyls derived from alkanes. m/z 43.018 $C_2H_2OH^+$ was identified as a fragment of acetic acid from laboratory tests.

Glycolaldehyde (m/z 61.028 $C_2H_4O_2H^+$) and hydroxyacetone (m/z 75.044 $C_3H_6O_2H^+$) could be interferences to acetic acid and propionic acid, respectively. However, these compounds have been mainly reported in environments affected by biogenic emissions and biomass burning, and acetic acid has been shown to be the dominant contributor to m/z 61.028 $C_2H_4O_2H^+$ in several environments (Karl et al., 2007;Fu et al., 2008;Haase et al., 2012).

### 3.4.2 Aromatic oxidation products

Aromatic oxidation products are of particular interest, as they have been shown to be an important source of radicals that drive ozone formation in oil and gas producing regions (Edwards et al., 2014). From laboratory and chamber studies, expected aromatic oxidation products include various diketones, phenols and nitrophenols, benzaldehyde-type compounds (from toluene and larger aromatics), and furanones (Wagner et al., 2003;Bloss et al., 2005;Wyche et al., 2009;Yuan et al., 2016b). PTR-MS can detect phenols, benzaldehydes, and furanones (Müller et al., 2012); detection of some dicarbonyls, such as glyoxal and methylglyoxal, may be difficult due to fragmentation,


strong humidity dependence, and interference from other species (Pang et al., 2014;Stönner et al., 2017).

By far the most abundant aromatic oxidation product detected was m/z 107.049 $C_7H_6OH^+$,
benzaldehyde (max. 360 pptv). Phenol ($C_6H_6O$), cresol ($C_7H_8O$), and methylfuranone ($C_5H_6O_2$) were detected at only an estimated 30-40 pptv maximum enhancement. Cresols, diketones, and furanones are expected to have a much higher yield at the observed $NO_x$ concentrations (average 1.1 ppb) (Smith et al., 1998;Müller et al., 2012), but are also much more reactive (Bierbach et al., 1994;Atkinson and Arey, 2003).

Müller et al. (2012) reported several unidentified masses resulting from chamber oxidation of trimethylbenzenes: m/z 87.044 $C_4H_6O_2H^+$ and m/z 101.060 $C_5H_8O_2H^+$. These masses were also detected during the April 23 SONGNEX flight, and they were quite significant relative to other oxygenates- m/z 87.044 at approximately 10% of the signal intensity of acetic acid. We classify these as aromatic oxidation products but do not have enough information to suggest a structure.

### 3.4.3 Cycloalkane oxidation products

The $H_3O^+$ ToF-CIMS detected several potential cycloalkane oxidation products as shown in Figure 11. The 2-double bond equivalent, 1-oxygen homologous series ($C_nH_{2n-2}OH^+$) does not decrease monotonically with carbon number: the species with four carbons (m/z 71.049 $C_4H_6OH^+$) and 6 carbons (m/z 99.080 $C_6H_{10}OH^+$) are the most abundant (Fig. 11a). In addition, m/z 71.049 $C_4H_6OH^+$ has a somewhat different spatial distribution than other oxygenates, including much higher intensity in the central part and sections of the northern part of the field, and may have a primary source (Section 3.4.4). The enhancement of the C6 oxygenate points to cycloalkane precursors. Similar ion masses were measured by the PTR-ToF-MS instrument deployed in the Uintah Basin, UT, in 2013 (Fig. 11b).

The smallest cycloalkane that exists in significant amounts in fossil fuels is cyclopentane (C5), but we might expect the C6 products to be more abundant because C6 precursors can be more abundant than C5 in oilfields (Simpson et al., 2010;Gilman et al., 2013). Additionally, OH reacts faster with substituted cycloalkanes (starting at C-6, methylcyclopentane) than unsubstituted cycloalkanes, although this probably has a smaller effect on ambient composition than the emissions composition (Calvert et al., 2008). Figure 11 also shows the cycloalkane precursors in the Uintah Basin,





measured by GC techniques (Edwards et al., 2013;Edwards et al., 2014). The suggested oxygenated products have a similar distribution. Data are not available from iWAS/GC-MS to show a similar precursor distribution for the Permian flight.

605

The C6 cycloalkane oxidation product ($C_6H_{10}OH^+$) could be a ketone (methylcyclopentanone or cyclohexanone), or cyclopentyl aldehyde. Aldehyde ions often fragment in PTR-MS by loss of $H_2O$ (Buhr et al., 2002). The fragmentary product of $C_6H_{10}OH^+$ dehydration, m/z 81.070 $C_6H_8H^+$, was significantly enhanced and correlated with photochemical species. Similar ions (m/z 95.086 $C_7H_{10}H^+$, m/z 109.101 $C_8H_{12}H^+$, etc.) also correlated with photochemical species and may also be fragments of cycloalkyl aldehydes. These ion masses were also seen by the PTR-ToF-MS in Utah in 2013, and showed behavior consistent with photochemical species (Fig. SI 6).

### 3.4.4 PAN and reactive nitrogen tracers

It has been previously reported that PTR-MS detects peroxy acetyl nitrate (PAN) at the protonated parent mass m/z 122.008 $C_2H_3NO_5H^+$, at m/z 77.023 $C_2H_4O_3H^+$, and at m/z 45.992 $NO_2^+$. Similarly, peroxy propionyl nitrate (PPN) is detected at m/z 91.039 $C_3H_6O_3H^+$ (Hansel and Wisthaler, 2000;de Gouw and Warneke, 2007;Kaser et al., 2013). However, m/z 77.023 may include contributions from another species such as acetone water cluster or peroxyacetic acid, and m/z 45.992 is not expected to be universally specific to PAN (de Gouw and Warneke, 2007;Kaser et al., 2013).

During the April 23 SONGNEX flight over the Permian, there was no peak detected at m/z 122.008 $C_2H_3NO_5H^+$. m/z 77.023 $C_2H_4O_3H^+$ and 91.039 $C_3H_6O_3H^+$ were detected at moderate and low intensities (4 ncps and 2 ncps average enhancement, respectively) and m/z 45.992 $NO_2^+$ was one of the most abundant ions detected (28 ncps average enhancement, similar to the average enhancement of benzene). M/z 77.023, m/z 91.039, and m/z 45.992 are compared to several independent measurements of reactive nitrogen: PAN, PPN, $NO_y$, $NO_y$-$NO_x$, and ethyl and propyl alkyl nitrates in Figure SI 8. When looking at all the SONGNEX flights, the slope of m/z 45.992 $NO_2^+$ versus $NO_y$-$NO_x$ (which represents all $NO_x$ oxidation products, including PAN), and the slope of m/z 77.023 versus PAN, vary significantly between different environments. Figure 12 shows the relationship between m/z 77.023 and PAN. Comparison of m/z 45.992 $NO_2^+$ with PAN



is included in the supplemental information (Fig. SI 9). The slope of m/z 45.992 vs $NO_z$ depends on the composition of $NO_z$ (Fig. SI 10a). The range of slopes probably depends on the atmospheric variability of $NO_z$, and not instrument conditions: the slope of m/z 77.023 vs PAN is not dependent

on instrument conditions such as drift tube humidity (Fig. SI 10b), which is consistent with behavior reported by Hansel and Wisthaler (2000).

These measurements suggest that these product ions cannot be attributed to PAN-type compounds only. Product ions at m/z 45.992 $NO_2^+$ almost certainly derive from a number of $NO_z$ species, with

a range of response factors; and there are probably at least two species that contribute to m/z 77.023 $C_2H_4O_3H^+$.

### 3.5 Other compounds with heteroatoms

VOC emissions from oil and gas operations are distinctly different from other commonly studied sources (urban areas, forests) with respect to the presence of non-photochemical species containing

nitrogen, sulfur, and oxygen heteroatoms. In this section we discuss cyclic nitrogen-containing species, $H_2S$, and two oxygenated species: methanol and m/z 71.049 $C_4H_6OH^+$.

### 3.5.1 Cyclic organic nitrogen species

An especially interesting observation is the broad enhancement of several nitrogen-containing organic species during the Permian flights. The most clearly enhanced ion, m/z 70.065 $C_4H_7NH^+$,

has signal intensity approximately 15% that of m/z 93.070 $C_7H_8H^+$, toluene (Fig. 13).

$C_4H_7N$ has several possible isomers, each with two degrees of unsaturation: a cyclic alkene structure (pyrroline) and several non-cyclic structures (C4 nitriles). Using mass spectral context, we suggest a cyclic structure for this compound. Time series of the homologous series ($C_nH_{2n-1}NH^+$) including $C_4H_7NH^+$ are shown in Figure 13. The species having one, two, and three carbon

atoms are not enhanced above background, whereas species containing four or more carbon atoms are enhanced and correlated with aromatic compounds (Fig. 13, Fig. SI 11). A cyclic structure is the most likely explanation for this pattern. A similar argument can be made for the homologous series ($C_nH_{2n-3}NH^+$) having three degrees of unsaturation (pyrroles). The series with two

($C_nH_{2n+1}NH^+$, pyrrolidines) and four ($C_nH_{2n-5}NH^+$, pyridines) degrees of unsaturation were not





significantly enhanced. Notably, m/z 84.081 $C_5H_9NH^+$ had a different distribution than other cyclic nitrogen compounds during the April 23 flight, with high mixing ratios in the eastern and western parts of the field, and overall correlated better with the photochemically produced species. We speculate that this mass may include a fragment of a photochemical product, analogous to the

interference with cycloalkanes discussed in section 3.2.2. Pyrroline (m/z 70.065) was enhanced by up to 200 pptv, and its concentration was typically comparable to C8 and C9 aromatics. Pyrrole (m/z 68.050) had a maximum enhancement of 40 pptv.

Organic nitrogen species detected over the Permian may have originated from the crude oil.

American shale oils can contain upwards of 2% nitrogen by weight, and a number of aromatic organic nitrogen species have been quantified in crude oil (Morandi and Jensen, 1966;Holmes and Thompson, 1983;Mushrush et al., 1999;Williams and Chishti, 2001). Porphyrins, nitrogen-containing geochemical fossils derived from chlorophyll, are known to exist in crude oil and contain both pyrrole and pyrroline as subunits (Fig. 14); aromatic nitrogen species may also form

through chemical processes in the oil reservoir. There are large differences in nitrogen content and speciation between reservoirs, which could explain why these species were detected in some regions and not in others (Baxby et al., 1994;Li et al., 1995;Oldenburg et al., 2007).

The reactivity of pyrroline is not known. We analyzed the relative decrease in $C_4H_7N$ in the

western half of the field as a function of OH exposure (see section 3.1), and estimated the rate constant with OH to be in the range of $(1.3-2.0)\times10^{-11}$ cm$^3$ molecule s$^{-1}$ with a best estimate of $1.7\times10^{-11}$ cm$^3$ molecule s$^{-1}$ (details in SI). Saturated nitriles have much slower reactivity with OH – reported values are around $2\times10^{-14}$ cm$^3$ molecule s$^{-1}$ (Harris et al., 1981;Atkinson et al., 2006) – so this high OH reactivity is further evidence that $C_4H_7N$ is not a nitrile. A similar analysis of

$C_4H_5N$ gave a rate constant of $2.5\times10^{-11}$ cm$^3$ molecule s$^{-1}$, which is much slower than the reported value for pyrrole ($1\times10^{-10}$ (Wallington, 1986)) but faster than butenenitrile ($1.4\times10^{-11}$ (Grosjean and Williams, 1992)). The pyrrole analysis is much less certain than that of pyrroline, due to the overall lower signal and higher noise. The rate constants for pyrroline and especially pyrrole could be significantly underestimated. In previous work, it has been shown that for fast-reacting species,

the derived rate constant was approximately constant and similar to the rate of the fastest-reacting aromatic used to calculate OH exposure (de Gouw et al., 2005). In our analysis, the fastest-reacting





aromatic considered was C10 aromatics, with a rate constant of $2.4 \times 10^{-11}$ cm$^3$ molecule s$^{-1}$. This may explain why the derived rate constant for pyrrole was lower than expected.

Heterocycles are highly reactive with nitrate radicals, especially pyrrole (k=$4.6 \times 10^{-11}$ cm$^3$ molecule$^{-1}$ s$^{-1}$) (Atkinson et al., 1985;Cabañas et al., 2004). If pyrroline is similarly reactive, then nitrogen heterocycles could potentially dominate VOC nitrate reactivity in oil and gas fields, because most other VOCs measured during SONGNEX (aromatics and aliphatics) have low reaction rates with nitrate radicals (k~$10^{-18}$-$10^{-15}$ cm$^3$ molecule$^{-1}$ s$^{-1}$). A comparison of nitrate

reactivity for species measured during SONGNEX, and a comparison to nitrate loss rates reported in the literature, are given in the supplementary information (Fig. SI 12, Section SI 4). We looked for species with two nitrogen atoms, but at our instrument resolution, they are extremely difficult to separate from isobaric hydrocarbon species unless they have very high signal intensity.

### 3.5.2 Hydrogen sulfide

Measurement of H$_2$S with PTR-MS has been described by Li et al. (2014). The H$_3$O$^+$ ToF-CIMS improves on the instrument described by Li et al. (2014) as the high mass resolution avoids the isobaric background interference from isotopes of methanol and HO$_2^+$. H$_2$S is detected at m/z 34.995 H$_2$SH$^+$ and was calibrated directly using a standard cylinder.

A comparison with the Picarro H$_2$S instrument is shown in Figure 15. There is good quantitative agreement between the two measurements. Compared to the Picarro instrument, the H$_3$O$^+$ ToF-CIMS H$_2$S measurement is more precise, and the 1s data are simultaneous with other H$_3$O$^+$ ToF-CIMS measurements, allowing easy comparison. The H$_3$O$^+$ ToF-CIMS H$_2$S measurement had a 3σ detection limit of 2.3 ppbv for a 1-s measurement (or 0.8 ppbv over the 6s measurement period

of the Picarro instrument).

H$_2$S had a maximum concentration of 12.6 ppb during the April 23 Permian flight, comparable to butanes. No other sulfur-containing species measurable with the H$_3$O$^+$ ToF-CIMS were enhanced above an estimated 1s 3σ detection limit of 30 pptv. (A few sulfur-containing species, such as m/z

121.032 C$_4$H$_8$SO$_2$H$^+$, have non-zero intensity but are instrument contaminants).



### 3.5.3 Methanol

Methanol was the most abundant VOC detected by $H_3O^+$ ToF-CIMS during the April 23 Permian flight. In this section we discuss some possible sources.

Methanol is used by the oil and gas industry. Significant primary emissions, especially from produced water storage infrastructure and storage containers on wellpads, have been measured in the Uintah Basin (Warneke et al., 2014;Mansfield et al., 2016). Industry uses of methanol include addition at well heads or further downstream in pipelines to prevent methane hydrate formation (Anderson and Prausnitz, 1986), to inhibit corrosion and scaling, as a lubricant, and as a solvent
in other applications (Mansfield et al., 2016). A more detailed investigation of methanol sources is outside the scope of this paper.

Globally, the dominant net source of methanol is direct biogenic emission, although there is a substantial and poorly constrained source from secondary production and oceans (Jacob et al.,
2005;Millet et al., 2008). Primary biogenic emissions can explain the high mixing ratios over the Haynesville region, but not over the Permian, given the absence of other biogenic VOCs.

Lewis et al. (2005) calculated rates of photochemical production of methanol from a set of VOC precursors, the most important of which were methane, iso-butane, iso-pentane, and acetaldehyde.
The magnitude of photochemical methanol production in the Permian was estimated by scaling the Lewis *et al.* precursor concentrations to the highest observed Permian VOC concentrations during the April 23 flight. Only 1-2 ppb of methanol would have formed after 2-3 days of aging – a small amount compared to the measured average 6 ppb and maximum 19 ppb. It should be noted that the Lewis et al. (2005) calculations were in the remote marine boundary layer and the
photochemical production rate in the Permian could be different. Methanol does have a much stronger correlation with photochemical species than with primary aromatic species (Fig. 7), but it also has a relatively long atmospheric lifetime.





### 3.5.4 M/z 71.049 C₄H₆OH⁺

M/z 71.049 $C_4H_6OH^+$ is typically interpreted as the sum of methyl vinyl ketone (MVK) and
methacrolein, two carbonyl products of isoprene oxidation (de Gouw and Warneke, 2007), but
isoprene was too low to produce the measured amount of $C_4H_6OH^+$ in the Permian Basin.
$C_4H_6OH^+$ had a spatial distribution that differed from photochemically produced species, including
a much larger enhancement in the central part of the flight and several north-south oriented plumes
running the length of the surveyed area (Fig. 2). Additionally, $C_4H_6OH^+$ was enhanced above what
might be expected from the distribution of precursor species and the enhancements of similar
oxygenates (Fig. 11). The enhancement in the more aged area of the field suggests that a large part
of this ion signal is generated by a photochemical species, but there may be other contributions.
Using the $H_3O^+$ ToF-CIMS average sensitivity factor for methyl vinyl ketone (MVK) and
methacrolein, the maximum boundary-layer enhancement of this species was 540 pptv.

The other carbonyl isomer, crotonaldehyde, has been reported in biomass burning emissions (Karl
et al., 2007), which were not evident during this flight. It is possible that MVK, methacrolein, or
crotonaldehyde could be directly emitted by anthropogenic sources, or photochemically derived
from a non-biogenic species. The cyclic isomer, dihydrofuran, is the oxygenated analogue of the
nitrogen heterocycle (pyrroline) discussed in section 3.4.1. Dihydrofurans are known to be
products of OH oxidation of alkanes (Lim and Ziemann, 2005).

Preliminary iWAS/GC-MS measurements show strong correlation between MVK, methacrolein,
and m/z 71.049 $C_4H_6OH^+$. The appropriate mass for dihydrofuran was not included in the selected-
ion-scan window of the iWAS GC quadrupole MS. Other evidence is needed to identify the
predominant isomer(s), and determine if there is a significant interference to PTR-MS
measurements of biogenic MVK and methacrolein in oil and gas producing regions.

## 4. Conclusions

We have analyzed PTR-ToF-MS mass spectra from aircraft measurements over several US oil and
gas producing regions. Our analysis is supported by comparison to independent co-deployed
instrumentation. We present a comparison between nine oil and gas basins of mixing ratios of



aromatics, major secondary species, methanol, and hydrogen sulfide. In every basin, measurements from $H_3O^+$ ToF-CIMS were dominated by small oxygenated compounds, especially C2-C4 photochemical products and methanol. Significant classes of hydrocarbon compounds

detected included aromatics, cycloalkanes, and alkanes. The $H_3O^+$ ToF-CIMS measurements of aromatics, methanol, and $H_2S$ agreed with independent measurements while methylcyclohexane and reactive nitrogen differed from independent measurement.

Between basins, there was large variation in the observed mixing ratios of aromatics, $H_2S$, and

methanol. In every basin, narrow, highly concentrated plumes with high mixing ratios of aromatics were measured, and average mixing ratios in many basins were comparable to concentrations observed downwind of large metropolitan areas. However, the profile of aromatics is different from that in urban air. We demonstrated the ability of $H_3O^+$ ToF-CIMS to detect hydrogen sulfide, and measured significantly enhanced $H_2S$ in the Permian and Haynesville regions. Methanol was

the single most abundant VOC observed by $H_3O^+$ ToF-CIMS and may have industrial sources. Compared to the variability in aromatics, methanol, and $H_2S$, photochemical compounds had similar mixing ratios in each basin. Additionally, the abundances of most oxygenates relative to acetone were similar between basins. This profile was also quite similar to that measured during a flight over the Los Angeles urban area during CALNEX 2010. The most highly variable compound

was acetic acid, which can include primary emission from agriculture, especially in the Denver-Julesburg Basin.

The Permian Basin had the highest overall mixing ratios of all species reported here. This region is the largest oil field in the United States but has not been studied extensively from an air quality

perspective. We conducted a detailed investigation of mass spectra recorded over Permian Basin. There are likely many chemically significant species, measureable by PTR-MS, in the atmosphere that are not currently routinely reported. This includes both primary species such as pyrroline and early-generation secondary species, such as the oxidation products of cycloalkanes. Pyrroline (m/z 70.065, $C_4H_7NH^+$) is especially interesting because it has not been previously reported as a VOC

associated with oil and gas emissions, and may account for a substantial fraction of nitrate reactivity. The $C_{10}H_{16}$ measurement, which is most likely adamantane or an unusual monoterpene, indicates the presence of larger (C10+) hydrocarbons emitted from oil and gas operations, which



are currently underexplored in the literature. The most important aromatic oxidation product detected was benzaldehyde; other products, including phenol and two unidentified oxygenates, were present at much smaller concentrations. Several ion masses that could be cycloalkane oxidation products were detected. Finally, we report several new interpretations of PTR-MS ion masses previously described in the literature.

**Acknowledgments**

A. Koss acknowledges funding from the NSF Graduate Fellowship Program. We thank the NOAA Aircraft Operations Center for their support with instrument installation on the NOAA WP-3D, research flights, and meteorological and aircraft data. We thank Ralf Staebler (Environment and Climate Change Canada) for the use of the Picarro $H_2S$ instrument. We thank Andy Neuman for his scientific advice and thoughtful comments on the manuscript. T. Hanisco, G. Wolfe, J. St. Clair, M. Thayer, and F. Keutsch acknowledge NASA-GeoCAPE award number NNX15AH83G for funding.





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



**Table 1.** SONGNEX chemical instrumentation

| Name of instrument | Species measured | Measurement technique | Citation or details |
|---|---|---|---|
| iWAS/GC-MS (improved Whole Air Sampler) | C2-C6 alkanes<br>Select cycloalkanes<br>C6-C8 aromatics<br>Small alkyl nitrates<br>Isoprene<br>Monoterpenes<br>Ethene<br>Ethyne<br>Methanol | Whole-air canister sampling followed by offline GC-MS analysis | (Lerner et al., 2017) |
| Aerodyne $C_2H_6$ instrument | Ethane | Tunable infrared laser direct absorption spectroscopy | (Yacovitch et al., 2014) |
| Picarro $CO_2/CH_4$ instrument | Methane<br>$CO_2$ | Cavity ring-down spectroscopy | (Peischl et al., 2012) |
| PAN CIMS | Peroxyacyl nitrates:<br>PAN<br>PPN<br>MPAN<br>APAN | Thermal dissociation/ $I^-$ chemical ionization mass spectrometry | (Slusher et al., 2004) |
| In-Situ Airborne Formaldehyde (ISAF) | Formaldehyde | Laser-induced fluorescence | (Cazorla et al., 2015) |
| Picarro G1204 $H_2S$ instrument | $H_2S$ | Cavity ring-down spectroscopy | Manufacturer specifications:<br>10 ppb 5-s LoD ($1\sigma$)<br>~5 second measurement interval<br>May experience interference from organics |
| NO/NO$_2$/NO$_y$ Chemi-luminescence | NO<br>$NO_2$<br>$NO_y$ | NO by NO/$O_3$ chemiluminescence<br>$NO_2$ by photolysis and NO/ $O_3$ chemiluminescence<br>$NO_y$ by Au converter and NO/$O_3$ chemiluminescence | (Ryerson et al., 2000;Pollack et al., 2010) |
| HNO$_3$/HCOOH CIMS | Nitric acid<br>Formic acid | $I^-$ chemical ionization mass spectrometry | (Neuman et al., 2016) |
| NOx CaRDS | NO<br>$NO_2$<br>$NO_y$ | Cavity ring-down spectroscopy | (Wagner et al., 2011) |


**Table 2.** Significant ion masses and interpretation. Mixing ratio (ppbv) enhancements are listed only for non-fragmentary ions of relatively certain identification. Ion masses with VOC interpretation of particular interest are highlighted using bold text.

| Ion Exact Mass (Th) | Ion formula | Previously reported interpretations | Interpretation in oil and gas producing regions | Max boundary layer enhancement (ppbv) during April 23 flight |
|---|---|---|---|---|
| 33.0335 | $CH_4OH^+$ | Methanol (de Gouw and Warneke, 2007;Blake et al., 2009) | Methanol | 19.09 |
| 34.9950 | $H_2SH^+$ | Hydrogen sulfide (Li et al., 2014) | Hydrogen sulfide | 12.6 |
| 41.0386 | $C_3H_5^+$ | General alkane/VOC fragment (Gueneron et al., 2015) MBO fragment (Kim et al., 2010); propyne (Stockwell et al., 2015) | General alkane/VOC fragment | |
| 43.0178 | $C_2H_2OH^+$ | Biogenic aldehyde fragment (Kim et al., 2010;Ruuskanen et al., 2011) Acetic acid fragment (de Gouw et al., 2003;Müller et al., 2012) | Fragment of acetic acid | |
| 43.0542 | $C_3H_7^+$ | General alkane/VOC fragment (Gueneron et al., 2015) Propene (Kuster et al., 2004;Knighton et al., 2012;Stockwell et al., 2015) | General alkane/VOC fragment | |
| 45.0335 | $C_2H_4OH^+$ | Acetaldehyde (de Gouw and Warneke, 2007;Blake et al., 2009) | Acetaldehyde | 3.60 |
| **45.9924** | **$NO_2^+$** | **PAN** (de Gouw et al., 2003;Müller et al., 2012;Kaser et al., 2013) | **Unresolvable $NO_z$ species** | |
| 57.0699 | $C_4H_9^+$ | General alkane/VOC fragment (Gueneron et al., 2015) Butenes (Karl et al., 2003) | General alkane/VOC fragment | |
| 59.0491 | $C_3H_6OH^+$ | Acetone (de Gouw and Warneke, 2007;Blake et al., 2009) | Acetone (negligible contribution from propanal) | 6.17 |
| 61.0284 | $C_2H_4O_2H^+$ | Acetic acid (de Gouw et al., 2003) | Acetic acid | 1.57 |
| **68.0495** | **$C_4H_5NH^+$** | **Pyrrole** (Brilli et al., 2014;Stockwell et al., 2015) | **Pyrrole** | **0.04** |
| **69.0699** | **$C_5H_9^+$** | **Isoprene** (Blake et al., 2009) **Cycloalkane fragment** (Gueneron et al., 2015) **MBO fragment** (Kim et al., 2010) | **Cycloalkane fragment + secondary species fragment** | |
| **70.0651** | **$C_4H_7NH^+$** | **Butane nitrile** (Brilli et al., 2014) | **Pyrroline (dihydropyrrole)** | **0.21** |
| **71.0491** | **$C_4H_6OH^+$** | **Biogenic MVK/methacrolein** (Blake et al., 2009) | **MVK/methacrolein /dihydrofuran** | **0.54** |
| 71.0855 | $C_5H_{11}^+$ | General alkane/VOC fragment (Yuan et al., 2014;Gueneron et al., 2015) | General alkane/VOC fragment | |
| 73.0648 | $C_4H_8OH^+$ | 2-butanone (MEK) (de Gouw and Warneke, 2007;Blake et al., 2009) | 2-butanone (MEK) (negligible contribution from butanals) | 1.96 |
| 75.0441 | $C_3H_6O_2H^+$ | Propionic acid (Ngwabie et al., 2008;Feilberg et al., 2015) Hydroxyacetone (Kim et al., 2010) | Propionic acid | 0.48 |
| **77.0233** | **$C_2H_4O_3H^+$** | **PAN** (Hansel and Wisthaler, 2000) | **Unknown species + PAN** | |
| 79.0542 | $C_6H_6H^+$ | Benzene (de Gouw and Warneke, 2007;Blake et al., 2009) | Benzene | 4.54 |
| **81.0699** | **$C_6H_8H^+$** | **Monoterpene fragment** (Kim et al., 2010) **PAH fragment** (Gueneron et al., 2015) | **Cyclopentyl aldehyde fragment (tentative)** | |




| 83.0855 | $C_6H_{11}^+$ | **Methylcyclopentane** (Yuan et al., 2014;Gueneron et al., 2015) | **Methylcyclopentane + secondary species fragment** | |
| 85.0648 | $C_5H_8OH^+$ | | **Cyclopentanone (tentative)** | **0.11** |
| 87.0441 | $C_4H_6O_2H^+$ | Aromatic oxidation product (Müller et al., 2012) 2-3,butadione (Stockwell et al., 2015) | Aromatic oxidation product | |
| 87.0804 | $C_5H_{10}OH^+$ | C5 carbonyls (Fall et al., 2001) MBO (Kim et al., 2010;Fall et al., 2001) | C5 carbonyls | 0.34 |
| 93.0699 | $C_7H_8H^+$ | Toluene (de Gouw and Warneke, 2007;Blake et al., 2009) | Toluene | 1.69 |
| 95.0855 | $C_7H_{10}H^+$ | Terpene fragment (Kim et al., 2009;Kim et al., 2010) | C7 cycloaldehyde fragment (tentative) | |
| 97.1012 | $C_7H_{13}^+$ | **C7 cycloalkanes** (Yuan et al., 2014;Warneke et al., 2015;Gueneron et al., 2015) | **Methylcyclohexane + secondary species fragment** | |
| 99.0804 | $C_6H_{10}OH^+$ | **Hexenal** (Fall et al., 2001;Ruuskanen et al., 2011;Park et al., 2013) | **C6 cycloalkane oxidation products (tentative)** | **0.12** |
| 101.0597 | $C_5H_8O_2H^+$ | Aromatic oxidation product (Müller et al., 2012) | Aromatic oxidation product | |
| 101.0961 | $C_6H_{12}OH^+$ | Hexanal (Rinne et al., 2005;Brilli et al., 2014) | C6 carbonyls | 0.08 |
| 105.0699 | $C_8H_8H^+$ | Styrene (Kuster et al., 2004) | Styrene | 0.03 |
| 107.0491 | $C_7H_6OH^+$ | Benzaldehyde (de Gouw et al., 2003) | Benzaldehyde | 0.37 |
| 107.0855 | $C_8H_{10}H^+$ | C8 aromatics (de Gouw and Warneke, 2007;Blake et al., 2009) | C8 aromatics | 0.65 |
| 111.1168 | $C_8H_{15}^+$ | **C8 cycloalkanes** (Yuan et al., 2014;Warneke et al., 2015;Gueneron et al., 2015) | **C8 cycloalkanes + secondary species fragment** | |
| 113.0961 | $C_7H_{12}OH^+$ | Heptenal (Brilli et al., 2014) | Cycloalkane oxidation product (tentative) | 0.06 |
| 115.1117 | $C_7H_{14}OH^+$ | | C7 carbonyls | 0.03 |
| 121.1012 | $C_9H_{12}H^+$ | C9 aromatics (de Gouw and Warneke, 2007;Blake et al., 2009) | C9 aromatics | 0.25 |
| 125.1325 | $C_9H_{17}^+$ | **C9 cycloalkanes** (Yuan et al., 2014;Warneke et al., 2015;Gueneron et al., 2015) | **C9 cycloalkanes + secondary species fragment** | |
| 135.1168 | $C_{10}H_{14}H^+$ | C10 aromatics (de Gouw and Warneke, 2007;Blake et al., 2009) | C10 aromatics | 0.08 |
| 137.1325 | $C_{10}H_{16}H^+$ | **Monoterpenes** (de Gouw and Warneke, 2007;Blake et al., 2009) | **Adamantane or mystery monoterpene** | 0.09 |
| 151.1481 | $C_{11}H_{18}H^+$ | | **Methyl adamantane (tentative)** | |



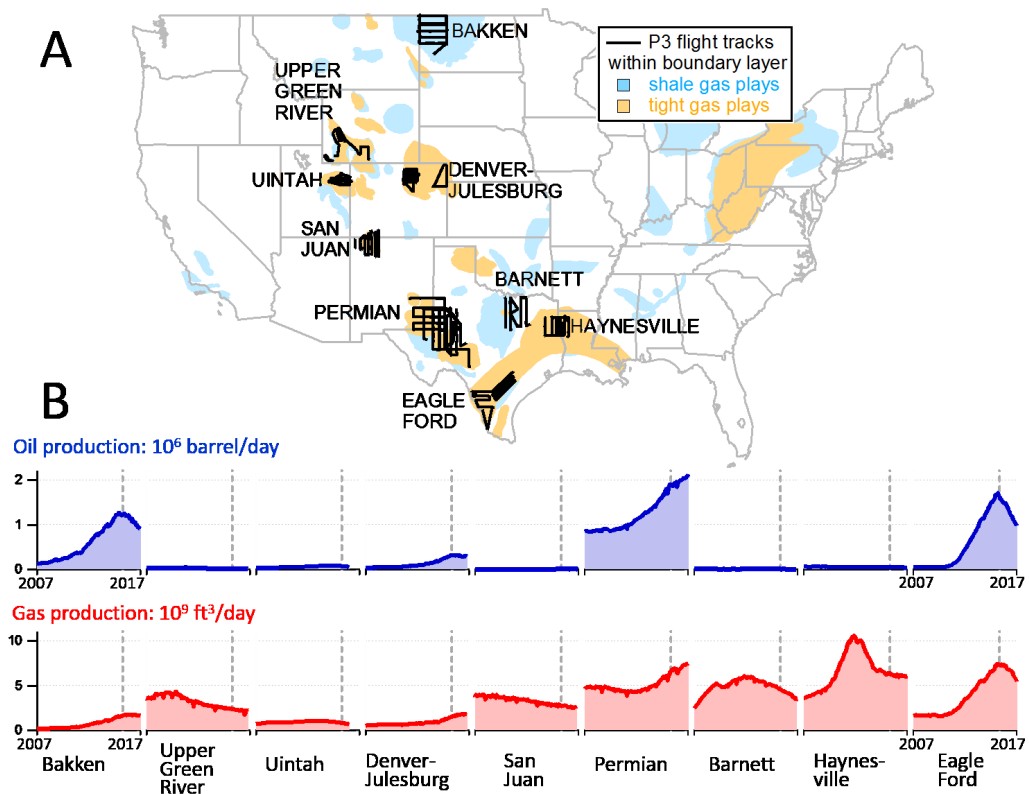

**Figure 1.** (a) SONGNEX study regions and P3 flight tracks. (b) Oil and natural gas production over time in various regions. The vertical dashed gray line marks the time period of the SONGNEX measurements. Production data from US Energy Information Administration (2017), Colorado Oil and Gas Conservation Commission (2017), State of New Mexico Oil Conservation Division (2017), Railroad Commission of Texas (2017), Wyoming Oil and Gas Conservation Commission (2017), and State of Utah Division of Oil Gas and Mining (2017).



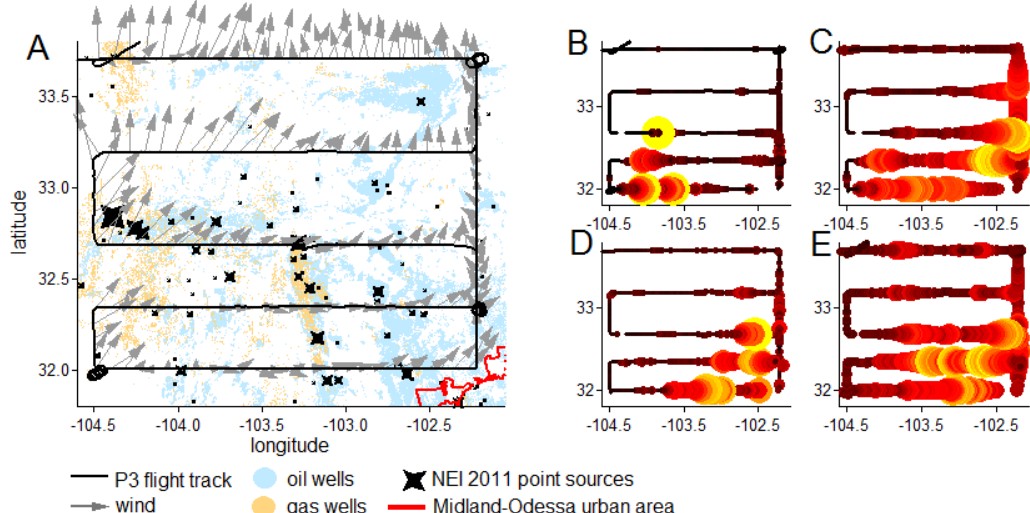

**Figure 2.** (a) April 23 flight track; wind direction; locations of oil and gas wells, NEI 2011 point sources, and the Midland/Odessa urban area. NEI point sources are sized by total reported VOC emission, ranging in the area shown from 0-400 ton/year. (b) Flight track colored and sized by toluene (0-1.7 ppbv). (c) Flight track colored and sized by acetaldehyde (0-3.6 ppbv). (d) Flight track colored and sized by $H_2S$ (0-8 ppbv). (e) Flight track colored and sized by m/z 71.049 $C_4H_6OH^+$ (0-86 normalized counts per second).





**Figure 3.** Mixing ratios observed within the boundary layer during SONGNEX flights. (a) Observed mixing ratios of aromatics (note log scale). (b) Mixing ratios of acetone. Tropospheric





background is marked by dashed line (Singh et al., 1995). (c) Mixing ratios of $H_2S$. Note split axes to show highest mixing ratios. Estimated limits of detection for 1 second measurement (relevant to maximum mixing ratio) and 5-hour average (relevant to basin average) are marked as dashed lines. (d) Mixing ratios of methanol. Note split axes. Estimates of tropospheric background are within the shaded gray box (Heikes et al., 2002). (e) Average mixing ratios of methane. The shaded box marks typical concentrations measured upwind of oil and gas basins, from Peischl et al. (2015);Pétron et al. (2014). (f) Average mixing ratios of α+β pinene from iWAS/GC-MS. The SENEX and CalNex data are sum of monoterpenes from PTR-qMS. (g) Average wind speeds in the boundary layer. (h) Average mixing ratios of $NO_x$.

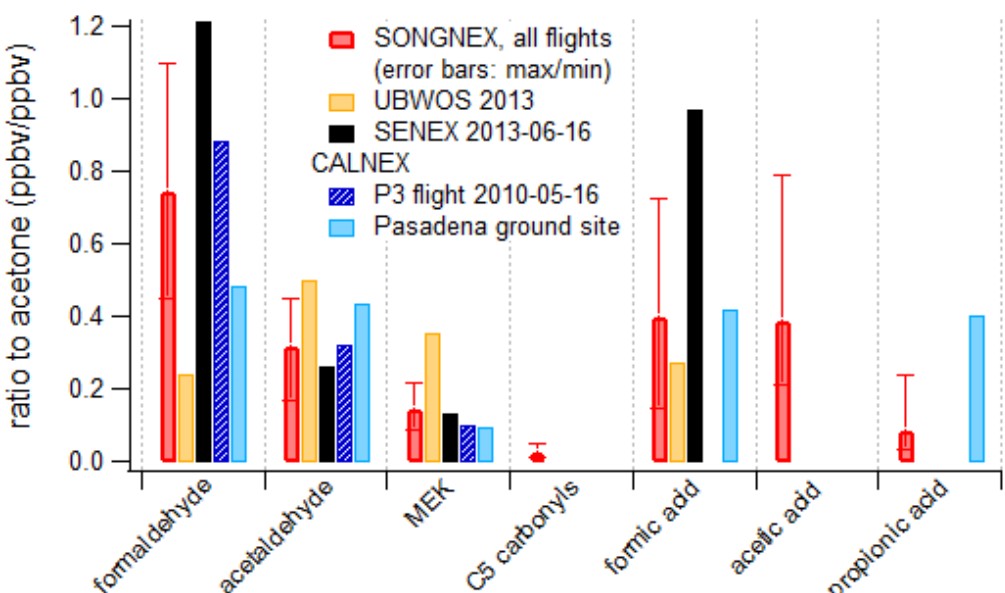

**Figure 4.** Average mixing ratios of oxygenates relative to acetone. Formaldehyde is from LIF instrument and formic acid from $HNO_3$/HCOOH CIMS. (For CalNex and SENEX instrumentation information, see https://www.esrl.noaa.gov/csd/field.html).



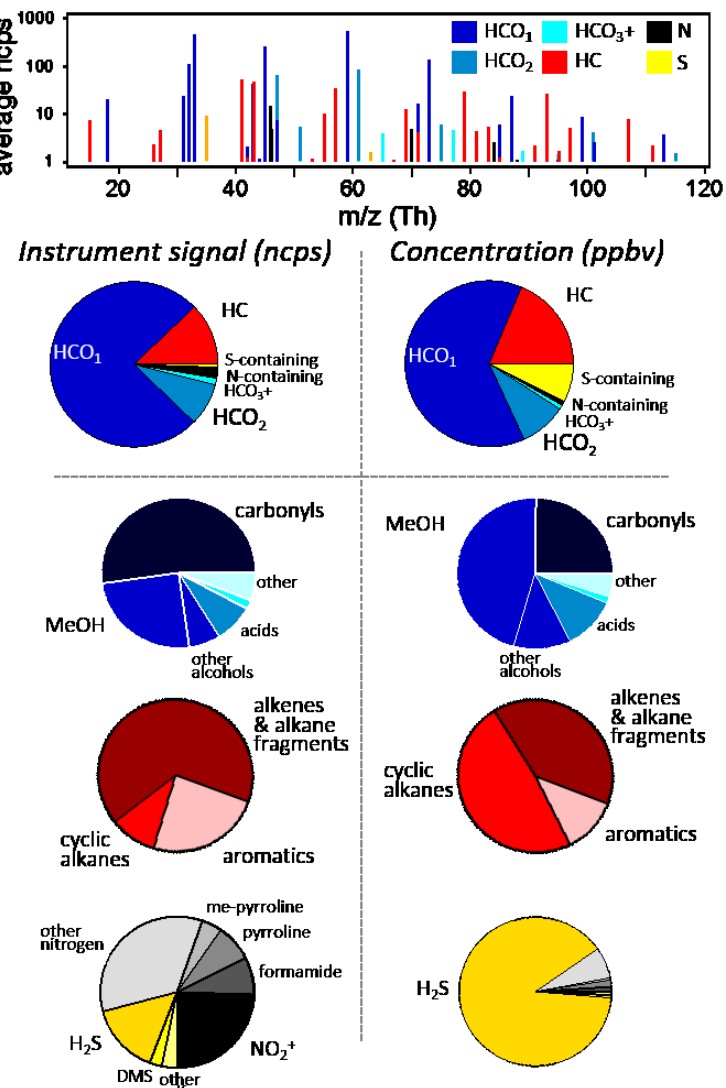

**Figure 5.** Overview of $H_3O^+$ ToF-CIMS mass spectra measured during the April 23 Permian flight. The categories "HC", "$HCO_1$", "$HCO_2$", etc. mean hydrocarbon species (without S or N) with no oxygen, with one oxygen atom, with two oxygen atoms, etc. Top: average mass spectrum, colored by elemental composition. Large pie chart: overall VOC composition. Color code is the same as mass spectrum. Small pie charts: composition of oxygenated (top), HC-only (middle), and N- and S- containing species (bottom). For all pie charts, the composition in terms of both relative instrument signal (ncps, left column) and mixing ratio (ppb, right column) is shown. See text for





a description of which species were included in the mixing ratio charts, and how concentrations were calculated.

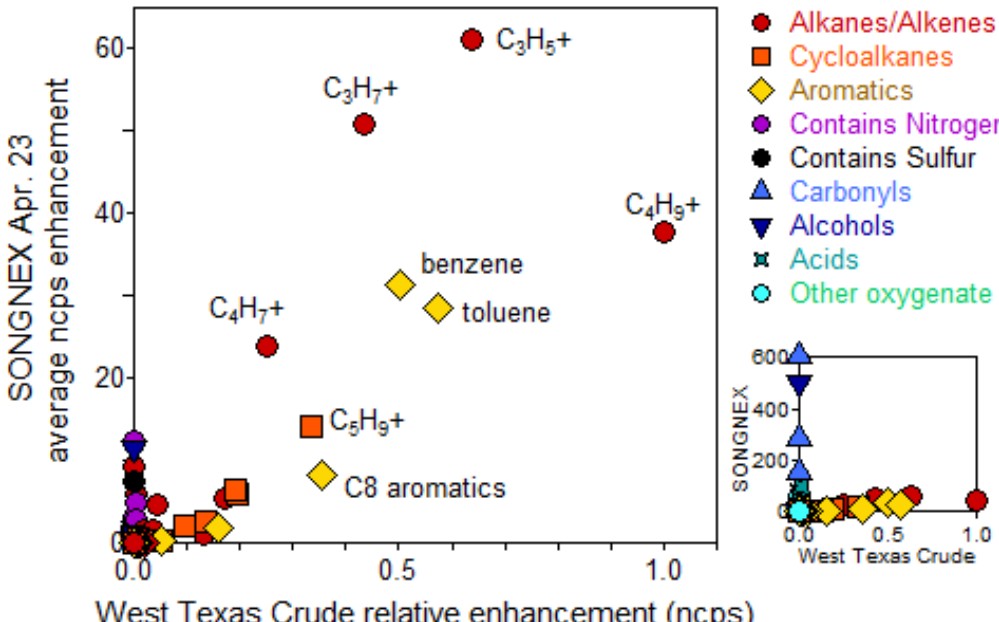

**Figure 6.** Comparison of $H_3O^+$ ToF-CIMS measurement of VOCs over the Permian Basin to VOCs evaporated from West Texas Crude oil. The values reported are the average boundary-layer enhancement in signal (ncps) during the SONGNEX April 23 flight, and the signal (ncps) relative to the most abundant mass (m/z 57.070 $C_4H_9^+$) in the West Texas Crude oil headspace.





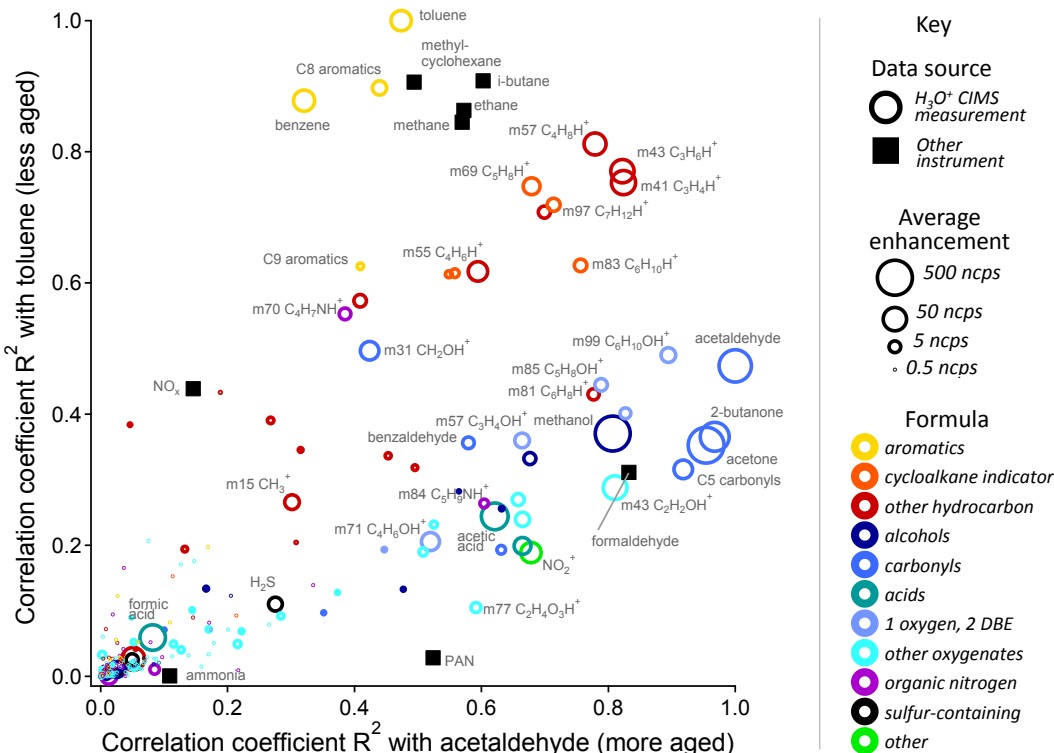

**Figure 7.** $H_3O^+$ ToF-CIMS VOC ion mass correlation with toluene, representative of less aged emissions, and with acetaldehyde, representative of more aged emissions. Toluene and acetaldehyde were selected for this analysis because they have high signal-to-noise ratio and unambiguous VOC ion interpretation.

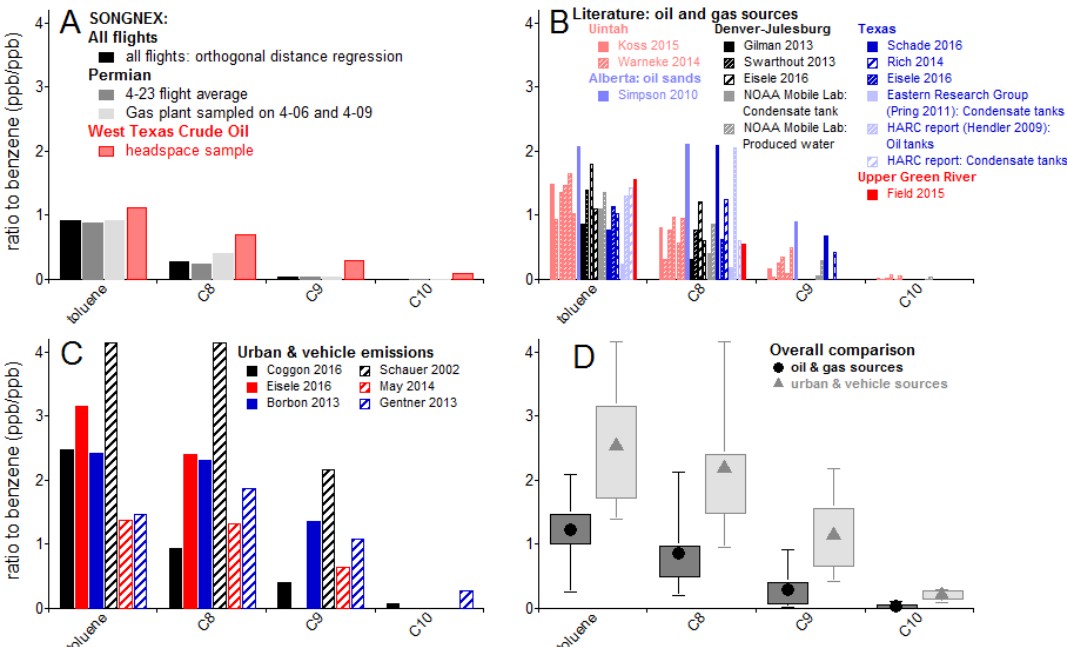

**Figure 8.** Comparison of C6-C10 aromatics distribution. Y-axis (molar ratio to benzene) is the same in each panel. (a) Measurements by $H_3O^+$ ToF-CIMS: SONGNEX flights and West Texas crude oil headspace. (b) Aromatic profiles from published studies of oil and gas fields, condensate tanks, oil tanks, and water storage tanks. (c) Aromatics profiles from published studies of urban areas and vehicle exhaust. (d) Average profiles of all oil and gas sources (panels a and b) and urban sources (panel c).





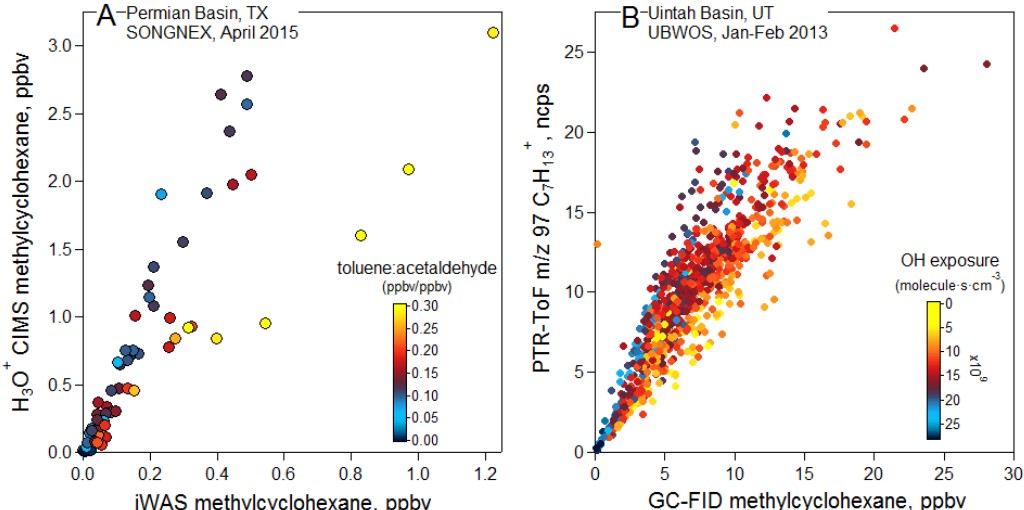

**Figure 9.** Comparison between PTR-ToF-MS and GC measurements of methylcyclohexane. A. SONGNEX 2015. B. UBWOS 2013.

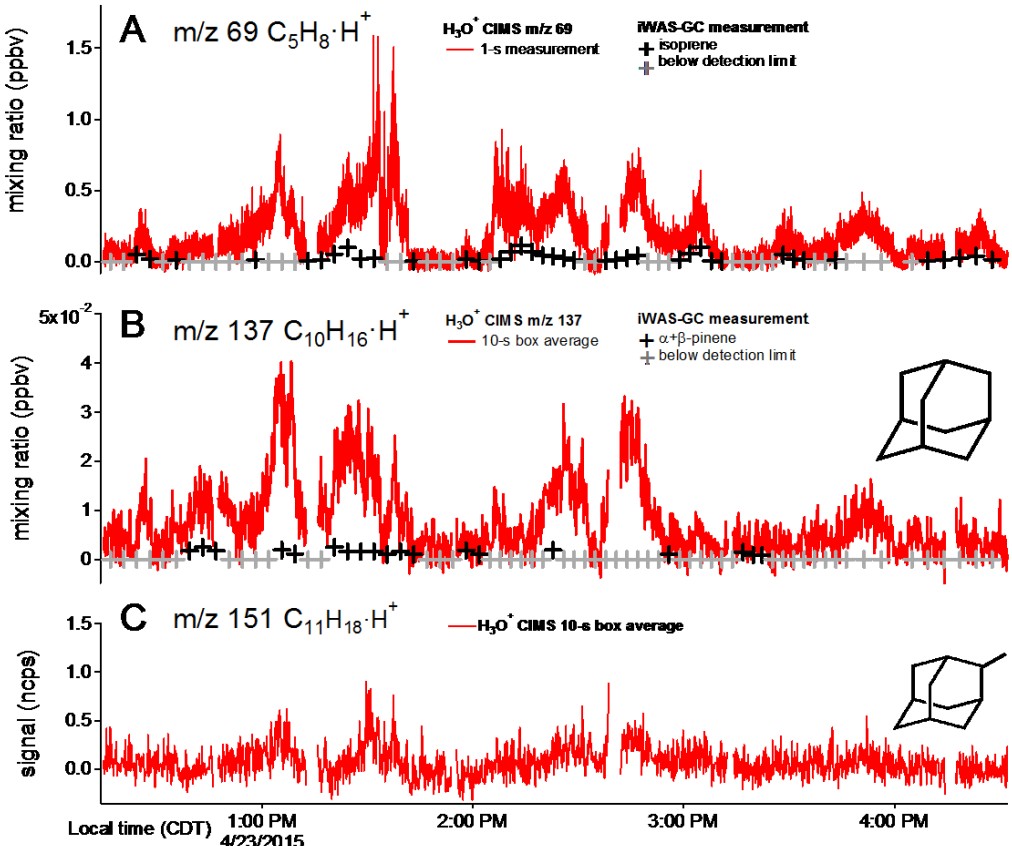

**Figure 10.** Time series of hydrocarbon masses. (a) m/z 69 $C_5H_9^+$, converted to ppb using isoprene sensitivity. (b) m/z 137 $C_{10}H_{16}H^+$, converted to ppb using monoterpenes sensitivity. (c) m/z 151 $C_{11}H_{18}H^+$, in ncps. The structures of adamantane and a methyl adamantane isomer (both isomers exist in oil) are included in panels (b) and (c), respectively.





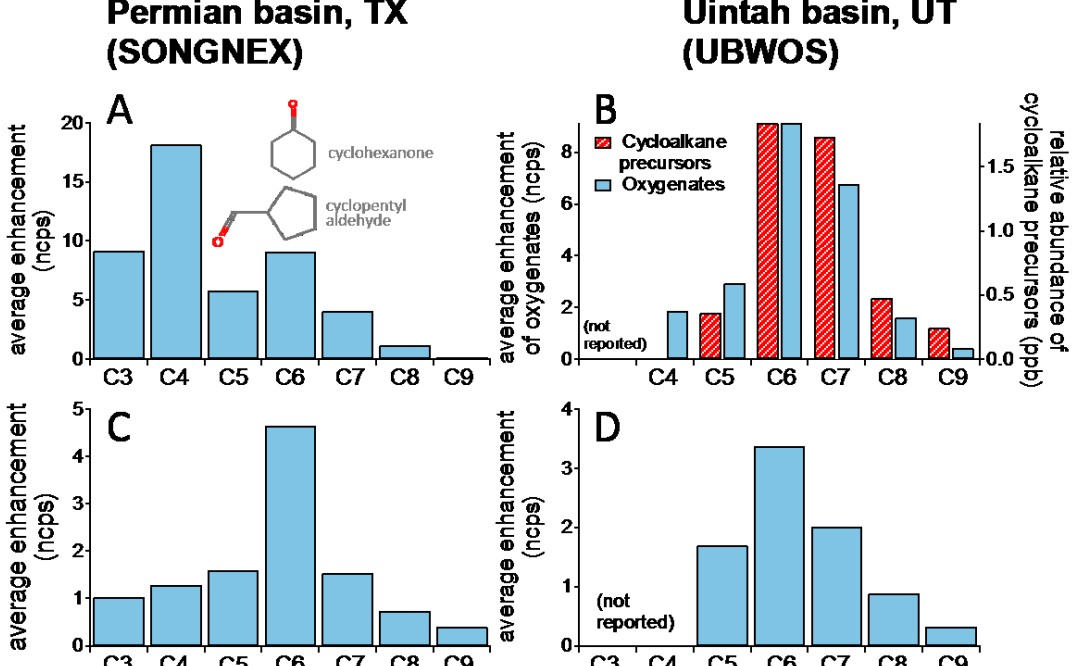

**Figure 11.** Ion masses related to cycloalkane oxidation products. The x-axis is the number of carbon in the molecule. (a) $C_xH_{2x-2}O$ oxygenates measured in the Permian basin during SONGNEX. (b) Cycloalkane precursors (measured by GC) and $C_xH_{2x-2}O$ oxygenates measured in the Uintah basin during UBWOS. Panels a and b show the same ion masses. (c) Possible dehydration masses ($C_xH_{2x-4}O$) of the oxygenates shown in (a), measured in the Permian basin. (d) Possible dehydration masses of the oxygenates shown in (b), measured in the Uintah basin.





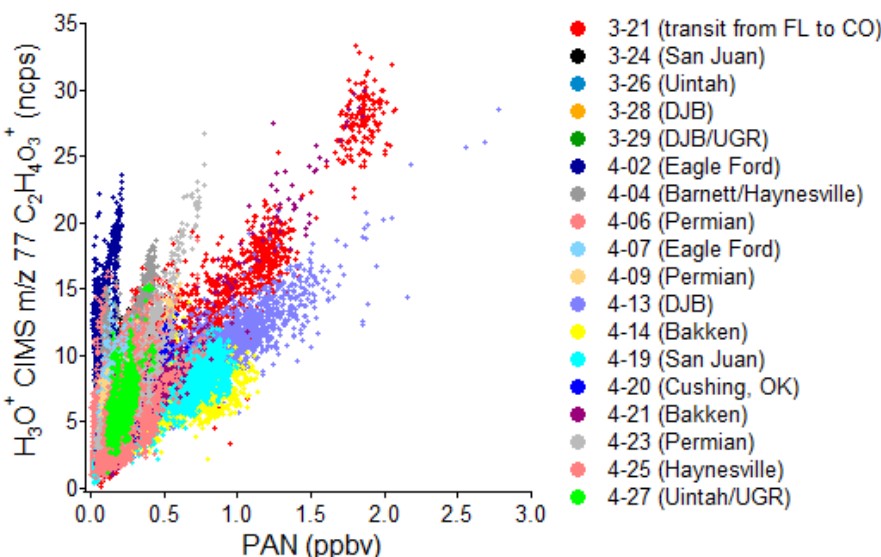

**Figure 12.** Relationship between m/z 77.023 $C_2H_4O_3H^+$ measured by $H_3O^+$ ToF-CIMS and PAN.

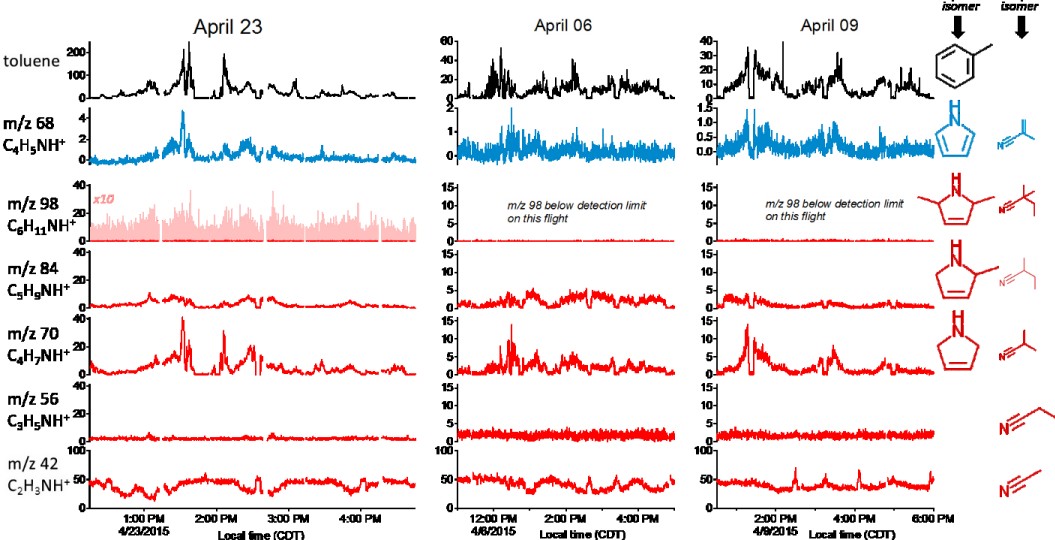

**Figure 13.** Time series of the homologous series of nitrogen-containing masses $C_nH_{2n-1}NH^+$ from n=2 to n=6 during the Permian basin flights on April 23, April 6, and April 9. Toluene and pyrrole are also shown. The y-axis is in units of normalized counts per second (instrument signal) for all ion masses (including toluene, $C_7H_8H^+$). Ten-second box averages of all species are shown for clarity. Data collected during vertical profiles are included in these time series, to show low



concentrations outside of the boundary layer. Right: some possible isomers for each mass. An example of a cyclic structure is given in the left column, and an example of a non-cyclic structure in the right column.

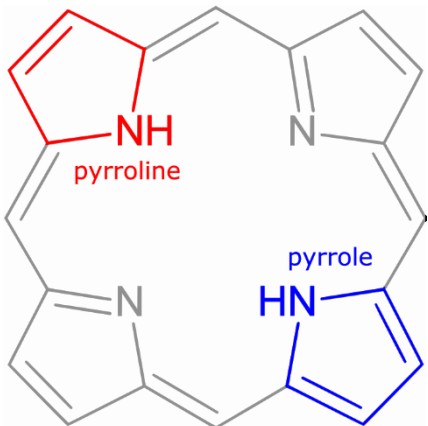

**Figure 14.** Structure of porphyrin with pyrrolic and pyrrolinic subunits highlighted.





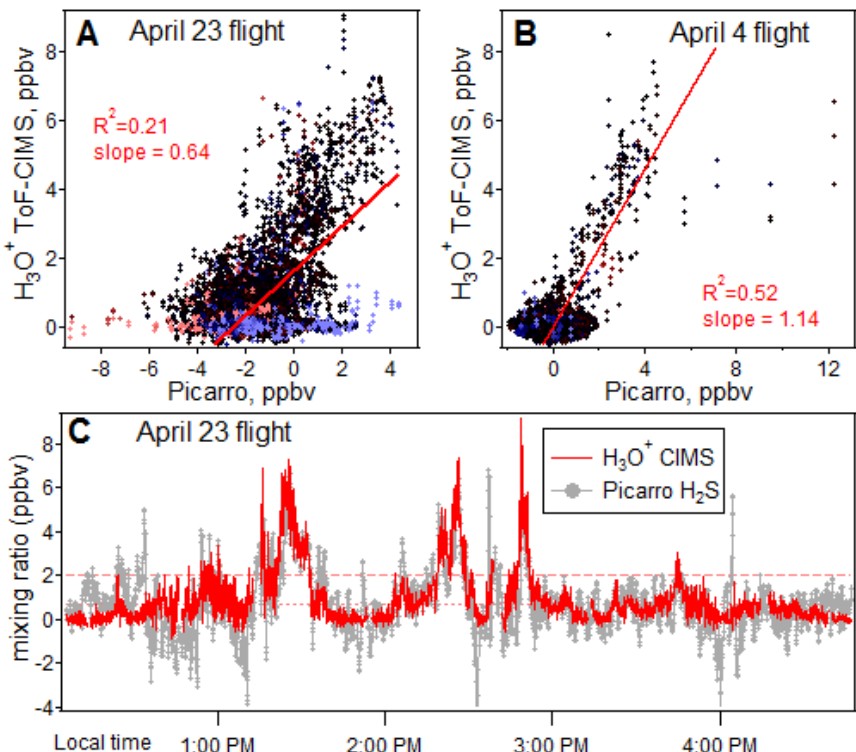

**Figure 15.** Scatterplot and time-series $H_2S$ comparison between $H_3O^+$ ToF-CIMS and Picarro CaRDS instruments. (a) Scatterplot of $H_3O^+$ ToF-CIMS vs Picarro $H_2S$ for April 23 flight (Permian). Data points when the aircraft was ascending (red) and descending (blue) are highlighted. (b) Scatterplot of $H_3O^+$ ToF-CIMS vs Picarro $H_2S$ for April 4 flight (Haynesville). (c) Time series during the April 23 flight. The $H_3O^+$ ToF-CIMS measurement has been averaged to a 6 second time basis. The Picarro $H_2S$ is offset by 2 ppb (the intercept in panel a). The 1-s and 6-s detection limits for the $H_3O^+$ ToF-CIMS are shown by dashed red lines.