# Peer review of "Observations of VOC emissions and photochemical products over US oil- and gas-producing regions using high-resolution H3O+ CIMS (PTR-ToF-MS)"

_Atmospheric Measurement Techniques, 2017_

## Referee Comment (RC1) · Anonymous Referee #1 · 8 Jun 2017

Koss et al. present a comprehensive data analysis of their series of airborne observations over US oil- and gas- producing regions. Since the ToF technology introduced to PTR applications, it has been highlighted as a main technological breakthrough to expand analytes to be quantified by taking advantage of high mass resolution. However, as far as I can tell, there has not been much of studies to comprehensively examine a wide swath of detected compounds. In this sense, Koss et al. present a very well-motivated study. The result and discussion section is quite lengthy for a good reason. It goes over a number of un-identified or under studied peaks on PTR-ToF-MS spectra

and discuss comprehensive details on its potential sources and artifacts. I strongly believe that the findings and discussion in the manuscript will be greatly beneficial to the PTR-ToF-MS user community. I have few minor comments and suggestions on the manuscript as described below.

1) It would be beneficial to include correlation plots for the chemical species overlapping PTR-ToF-MS and Whole Air-GC datasets especially species such as benzene and toluene.

2) Page 15 Line 457 – 464: I know that this manuscript tries to describe mainly technical aspects but it would be helpful to describe little further what 'photochemistry' is 'low' and 'high' means.

3) Page 19 Line 543: MEK is also known for a solvent so is there any possibility that it may come from direct emission?

---

## Short Comment (SC1) · 12 Jun 2017

The authors have presented data from PTR-TOF-MS from various oil and natural gas basins in the United States. The concentrations of various VOCs are discussed and compared to other studies/locations for context. The presence of some compounds in the Permian basin are investigated and identified. A lot of data were presented in this paper and it contains a host of information. The paper is well-written, concise, and flows well. The information presented is important to the scientific community investigating oil and gas emissions and those using PTR-TOF-MS. I recommend publication with a

few minor comments.

Comments: Figure 4: I find the error bars of the SONGNEX flights (max/min) confusing. Could the other clarify this please?

Line 730-731: Since this paper does not specifically look at methanol but presents an overview of the VOCs I think the authors can delete the "A more detailed.... this paper." sentence.

---

## Author Comment (AC1)

**We thank both reviewers for their positive comments. Below we respond to specific comments.**
Reviewer comments in black text.
**Author comments in bold blue text.**

**Response to Reviewer 1.**

Koss et al. present a comprehensive data analysis of their series of airborne observations over US oil- and gas- producing regions. Since the ToF technology introduced to PTR applications, it has been highlighted as a main technological breakthrough to expand analytes to be quantified by taking advantage of high mass resolution. However, as far as I can tell, there has not been much of studies to comprehensively examine a wide swath of detected compounds. In this sense, Koss et al. present a very well motivated study. The result and discussion section is quite lengthy for a good reason. It goes over a number of un-identified or under studied peaks on PTR-ToF-MS spectra and discuss comprehensive details on its potential sources and artifacts. I strongly believe that the findings and discussion in the manuscript will be greatly beneficial to the PTR-ToF-MS user community. I have few minor comments and suggestions on the manuscript as described below.

1) It would be beneficial to include correlation plots for the chemical species overlapping PTR-ToF-MS and Whole Air-GC datasets especially species such as benzene and toluene.

**We already published these graphs in Lerner et al. (2017) and Yuan et al. (2016), so we do not think it is appropriate to add them to the main text of this manuscript. However, we understand that the plots should be easily accessible. So, we have reproduced the plots in the Supplemental Information as Figure SI 1.**

2) Page 15 Line 457 – 464: I know that this manuscript tries to describe mainly technical aspects but it would be helpful to describe little further what 'photochemistry' is 'low' and 'high' means.

**We replaced "low" with "less active (daily ozone formation of 16 ppbv and ozone below 51 ppbv)" and "high" with "more active (daily ozone formation of 39 ppbv and 49 ozone exceedances)".**

3) Page 19 Line 543: MEK is also known for a solvent so is there any possibility that it may come from direct emission?

**This is an interesting question, because the use and emissions of solvents by the oil and gas industry is not particularly well known.**

**We observed during the April 23 Permian flight, MEK correlated well ($R^2>0.9$) with a number of other species that are typically produced by photochemistry, including acetaldehyde, acetone, and pentanones; and had a spatial distribution similar to formic acid (measured by HCOOH CIMS) and PAN (measured by PAN CIMS). Some of these other species, such as acetone, could also have primary solvent sources, but PAN probably does not. Additionally, the ratios of acetaldehyde and MEK to acetone are similar to those measured during the UBWOS 2013 campaign in Utah, where we know from box modeling that these species were produced mostly from photochemistry (Figure 4 in text). For these reasons we think these oxygenates are mostly photochemical products.**

**Unfortunately, we don't have a way to constrain the contribution from solvents during SONGNEX. During the Uintah Basin, UT Winter 2012 study we did mobile lab measurements close by oil and gas**

well pads and other facilities (Warneke et al., 2014). During these measurements we did not observe any MEK emissions and therefore did not discuss MEK in that paper.

In the text, at line 545, we have added, "Some of these compounds might also have direct sources from industrial solvent use, but during the April 23 flight the photochemical source was dominant, because they were only found enhanced over background in the areas, where peroxy acetyl nitrate (PAN) and other photochemical products were also elevated (Section 3.2)."

**Response to Reviewer #2**

The authors have presented data from PTR-TOF-MS from various oil and natural gas basins in the United States. The concentrations of various VOCs are discussed and compared to other studies/locations for context. The presence of some compounds in the Permian basin are investigated and identified. A lot of data were presented in this paper and it contains a host of information. The paper is well-written, concise, and flows well. The information presented is important to the scientific community investigating oil and gas emissions and those using PTR-TOF-MS. I recommend publication with a few minor comments.

Comments: Figure 4: I find the error bars of the SONGNEX flights (max/min) confusing. Could the other clarify this please?

We have added to the caption, "The whiskers on the PTR-ToF-MS measurement box plots (panels a-d) show the maximum and minimum (if above detection limit) concentrations measured across all SONGNEX flights."

Line 730-731: Since this paper does not specifically look at methanol but presents an overview of the VOCs I think the authors can delete the "A more detailed.... this paper." sentence.

Agreed. Deleted sentence.

**References**
Lerner, B. M., Gilman, J. B., Aikin, K. C., Atlas, E. L., Goldan, P. D., Graus, M., Hendershot, R., Isaacman-VanWertz, G. A., Koss, A., Kuster, W. C., Lueb, R. A., McLaughlin, R. J., Peischl, J., Sueper, D., Ryerson, T. B., Tokarek, T. W., Warneke, C., Yuan, B., and de Gouw, J. A.: An improved, automated whole air sampler and gas chromatography mass spectrometry analysis system for volatile organic compounds in the atmosphere, Atmos. Meas. Tech., 10, 291-313, 10.5194/amt-10-291-2017, 2017.
Warneke, C., Geiger, F., Edwards, P. M., Dube, W., Pétron, G., Kofler, J., Zahn, A., Brown, S. S., Graus, M., Gilman, J. B., Lerner, B. M., Peischl, J., Ryerson, T. B., de Gouw, J. A., and Roberts, J. M.: Volatile organic compound emissions from the oil and natural gas industry in the Uintah Basin, Utah: oil and gas well pad emissions compared to ambient air composition, Atmos. Chem. Phys., 14, 10977-10988, 10.5194/acp-14-10977-2014, 2014.
Yuan, B., Koss, A., Warneke, C., Gilman, J. B., Lerner, B. M., Stark, H., and de Gouw, J. A.: A high-resolution time-of-flight chemical ionization mass spectrometer utilizing hydronium ions (H3O+ ToF-CIMS) for measurements of volatile organic compounds in the atmosphere, Atmos. Meas. Tech., 9, 2735-2752, 10.5194/amt-9-2735-2016, 2016.